SPECIAL ISSUE
THE EXTRACELLULAR ENVIRONMENT

# TiFM2.0 – versatile mechanical measurement and actuation in live embryos

Ana R. Hernandez-Rodriguez[1,2,*], Yisha Lan[1,2,*], Fengtong Ji[1,2], Susannah B. P. McLaren[1,2], Joana M. N. Vidigueira[1,2], Ruoheng Li[1], Yixin Dai[1,3], Emily Holmes[1,2], Lauren D. Moon[1,2], Lakshmi Balasubramaniam[1,2] and Fengzhu Xiong[1,2,‡]

## ABSTRACT

During development, spatial-temporally patterned tissue-level stresses and mechanical properties create diverse tissue shapes. To understand the mechanics of small-scale embryonic tissues, precisely controlled sensors and actuators are needed. Previously, we reported a control-based approach named tissue force microscopy (TiFM1.0), which combines dynamic positioning and imaging of an inserted cantilever probe to directly measure and impose forces in early avian embryos. Here, we present an upgraded system (TiFM2.0) that uses interferometer positioning to minimise probe holder footprint, enhancing accessibility and imaging signal. This new design enables a double-probe configuration for bidirectional stretching, compression and stress propagation experiments. As proof-of-concept, we showcase a variety of examples of TiFM2.0 applications in chicken and zebrafish embryos, including the characterisation of mechanical heterogeneities important for the morphogenesis of the chicken posterior body axis. We also present simplified designs and protocols for the replication of TiFM systems with minimal custom engineering for developmental biology labs.

KEY WORDS: Tissue mechanics, Morphogenesis, Force, Rheological property, Embryo, Body axis

## INTRODUCTION

Multicellular organisms create tissue shapes throughout development by actively regulating mechanical forces and tissue material properties through patterned cell activities (Adams et al., 1990; Guillot and Lecuit, 2013; Mongera et al., 2019; Moon and Xiong, 2021; Stooke-Vaughan and Campàs, 2018; Sutlive et al., 2022). These mechanical factors also provide feedback to cell behaviours and genetic regulation, forming a close interplay that contributes to the robustness of morphogenesis (Chan et al., 2019; Eritano et al., 2020; Lu et al., 2024; Shyer et al., 2015; Xiong et al., 2020). Such interplays are deeply embedded in the evolution history of development and work across scales, leading to a high degree of complexity. Disentangling these complexities would require quantitative measurement and control of tissue mechanics, integrated with molecular and cellular approaches, such as -omics (van den Brink et al., 2020) and high-resolution live imaging (Xiong and Megason, 2015). These precision mechanical tools are also expected to impact engineering efforts that aim to recreate tissue and organs of functional shapes (Campàs, 2016).

Existing *in vivo* tissue mechanics approaches can be categorised largely by the presence or absence of direct contact with the sample. Non-contact methods include light and acoustic imaging of the sample and interpretation of the returned signal under certain assumptions, such as tissue geometry, composition and viscoelastic behaviours. The signals range from the expression and dynamics of key cytoskeleton-related molecules (Bertet et al., 2004; Chanet et al., 2017), their associated sensors (Riedl et al., 2008), cell shapes and movements (Merkel and Manning, 2017; Xiong et al., 2014), cell/tissue recoils triggered by laser ablation (Campinho et al., 2013) or optogenetics (Sampayo et al., 2023; Yamamoto et al., 2021), to scattering patterns and spectrum shifts (Dal Molin et al., 2015; Mulligan et al., 2016; Prevedel et al., 2019). These approaches offer accessibility with low-invasiveness and can cover large tissue areas. The disadvantages are that they often do not yield quantitatively definitive results as the measurements usually capture only a fraction of the factors that contribute to the tissue forces and properties, and assumptions used in data interpretation can strongly bias the conclusions. For example, while inferring junctional tension and pressure from modern high-resolution images provides a powerful way to map mechanics in 2D (Roffay et al., 2021) and recently 3D (Ichbiah et al., 2023) tissues, this method relies on assumptions such as equilibrium conditions and sheet-like or foam-like organisation of cells, which simplify yet limit the recoverable mechanical factors. On the other hand, contact methods use an inserted or injected sensor, usually with known and/or controllable mechanical properties, to directly detect its interactions with cells and tissues. An example of this kind of probe-based approach is the optical tweezer, which allows local junctional tension and cellular viscoelasticity to be directly measured (Bambardekar et al., 2015; Ferro et al., 2020). Scaling-up this principle to the tissue level, various probes have been developed and their *in situ* deformations and deflections along with tissue deformations directly imaged (Campàs et al., 2014; Chan et al., 2023; Dzementsei et al., 2022; Kato and Inomata, 2023; Serwane et al., 2017) or indirectly sensed (Wang et al., 2024). Whilst providing definitive measurements enabling the establishment of the regimes of forces and mechanical properties in which developing tissues operate, the results from these approaches must be interpreted carefully, considering the effective spatial-temporal resolution and scales – in particular,

[1]Gurdon Institute, University of Cambridge, Cambridge CB2 1QN, UK.
[2]Department of Physiology, Development and Neuroscience, University of Cambridge, Cambridge CB2 3DY, UK. [3]Life Science Institute, Zhejiang University, Hangzhou 310058, China.
*These authors contributed equally to this work

‡Author for correspondence (fx220@cam.ac.uk)

A.R.H.-R., 0000-0003-0246-846X; Y.L., 0000-0003-4675-4767; F.J., 0000-0002-9624-3503; S.B.P.M., 0000-0001-7495-0357; J.M.N.V., 0000-0002-3756-2854; L.B., 0000-0001-6881-8261; F.X., 0000-0002-6153-0254

whether the scales are relevant to tissue morphogenesis and how the invasiveness of sensors to the local tissue environment causes mechanical artefacts. Furthermore, considerable probe design and precision engineering expertise (Chan et al., 2023; Ghosh et al., 2018; Lee et al., 2017; McLaren and Xiong, 2024; Serwane et al., 2017) is often required in developing and deploying mechanical sensors for small and soft embryonic tissues, reducing the throughput and applicability of such systems for wide use in the developmental biology community.

The prototype TiFM system (TiFM1.0) (Chan et al., 2023) is a cantilever-based measurement and actuation tool that vertically inserts a soft atomic force microscope (AFM) probe into intact, live embryonic tissues. The probe deflects under the morphogenetic forces from the tissue or imposed forces by the motorised holder. Real-time, precise measurements of the positions of the probe tip and holder allow the force to be quantified and controlled dynamically. TiFM1.0 used charged capacitors to monitor the position of the cantilever holder, which results in a large probe head that limits sample illumination. The capacitors are also sensitive to the humidity in the environment of the embryo sample, where condensations could cause discharges that affect the measurements. Another challenge associated with limited access space on top of the sample is that further modifications, such as multiplexing of the probes, are not allowed. Here, we overcome these challenges by incorporating interferometer-based positioning, greatly reducing the probe head footprint. This new design allows for improved top illumination, a stage mounted incubation chamber, and a double-arm probe holder to be added. These modifications expand the applications possible to include different types of stress loading and rheological assessment, and enhanced coverage and reliability to provide a new generation of tissue force microscopy (TiFM2.0), suitable for studying a variety of tissue mechanics questions in live embryos.

## RESULTS

### Design and operation of the TiFM2.0 system

TiFM2.0 (Fig. 1A) offers a modular design, using a probe head containing either 1 or 2 electrical piezos in parallel, enabling independent and fine positioning of the probes along the x-axis through a voltage controller (Fig. 1Bi,Bii). The whole probe head is adjustable by a three-axis micromanipulator, while the second piezo has an additional two-axis (yz) micromanipulator for fine alignment of two probes relative to each other. The overall much smaller probe head construct (compared to TiFM1.0; Chan et al., 2023) can be lowered into an environmental well fitted with a controlled heating holder and a water supply micropipette, where standard glass-bottom sample dishes fit. To maximise the flexibility and range of interactions in the dual-probe setting, we designed the holder for AFM chips on an inclined surface (Fig. 1Bii). The inner surface (between two holders) has an angle of 11.3° to the vertical plane and the outer surface has a 20° inclination to the vertical plane. This allows the distance between two AFM chips to vary between a few hundred μm on the negative (i.e. the two probes crossing each other) to >1 mm, which provides a wide operating space for different tissue sizes and functions including stretching, compressing and twisting. The mounting surface of the holder supports a mirror for the interferometer lasers to measure probe holder positions from the sides. The probes and sample are imaged from the bottom side via a wide-field fluorescent microscope with a high frame-rate camera (Fig. 1C,D). Similar to TiFM1.0, the theoretical force sensitivity limit is at the order of 10 nN, set by the resolution of probe tip tracking currently at the order of 1 μm; the softest probes we tested at the order of 0.01 N m$^{-1}$. In practice, we detect most embryonic

tissue forces between 100 nN and 10 μN (tissues yield at the higher end). The micromanipulators can be further automated for programmable probe alignments, and the microscope can be changed to a high-resolution inverted confocal microscope or a fast oblique plane light-sheet microscope (OPM) (Dunsby, 2008; Sirinakis et al., 2024) depending on the applications required.

An experiment on TiFM follows a set-up sequence where probes are aligned with samples in 3D coordinates and a calibration run is performed where probe movement ($x_T$) together with the holder ($x_C$) without resistance is recorded by the microscope and the interferometer (Fig. 1Biii,E). Because of the rotational movement of the piezo and the difference of heights ($L_{CT}$, Fig. 1Biii) at which $x_T$ and $x_C$ are measured depending on the specific probe holder used, $x_T$ and $x_C$ change at different rates as a function of $L_{CT}$ (Fig. 1E; Fig. S1A,D). Next, a measurement sequence is performed, where samples are inserted and actuated with the probe(s) under the control of a user-defined programme, producing position data tracks of the holder ($x_{C1}$) and the probe inside the sample ($x_{T1}$) (Fig. 1Biii,F; Fig. S1B,E). The normal force on the probe along the x-axis at any time point is then given by $F=k\cos(\theta)[f(x_{C1})-x_{T1}]$ (Fig. 1G; Fig. S1C,F), and can be converted to stress as $\sigma=Fw^{-1}(z-z_0)^{-1}$, where $k$ is the spring constant, $w$ is the width of the probes, $z$ is the insertion depth, $z_0$ is the upper tissue surface and $\theta$ is the insertion angle relative to the z-axis. Note that under the embedded configuration of the probe-tissue interface, the detected force/stress on the probe has multiple sources of origin (such as compressive deformation along the x-axis and shear deformation along the y-axis near the probe edge). Therefore, while the results are indicative of the forces and material properties of the tissue location, appropriate models (e.g. finite element models; Agero et al., 2010; Michaut et al., 2025b preprint) are needed to quantitatively interpret them together with the strain field around the probe measured by tracking markers and features from the images, to reveal mechanical heterogeneities and anisotropy of the sample (further discussed in Discussion section).

### Local stretching and compression of small embryonic tissues

A key technical challenge in mechanically perturbing early embryonic tissues is to apply controlled, physiologically relevant forces and deformations. Taking advantage of the adhesion between the actuating tools and the connectiveness of epithelial tissues, recent works have allowed small tensions and shear forces to be applied in a minimally invasive manner in the avian embryo (Kunz et al., 2023; Michaut et al., 2025b preprint; Oikonomou et al., 2025). The TiFM system vertical probe-tissue interface provides the additional advantage of applying forces on bulk tissues including mesenchymal ones (Chan et al., 2023). In the example of the pre-somitic mesoderm (PSM), we performed local stretching with TiFM2.0 to compare the tissue responses in the anterior versus posterior PSM (i.e. aPSM versus pPSM), regions known to have a transition of cell state (Chal et al., 2017; Delfini et al., 2005) (mesenchymal posterior to epithelial anterior), cell density (Bénazéraf et al., 2017; Xiong et al., 2020) (low posterior to high anterior) and extracellular matrix (ECM) enrichment (Bénazéraf et al., 2010; Rifes and Thorsteinsdóttir, 2012) and distribution (Michaut et al., 2025a). A course of stepwise stretching along the anterior-posterior axis, followed by probe holding and relaxation by probe retrieval was performed in the mediolateral centres of the aPSM and pPSM (Fig. 2A; Movie 1). The tissues showed distinct responses and dynamics towards the final holding strain. In the aPSM, micro-tearing (Fig. 2A′, arrows, and Fig. 2A‴) appeared on

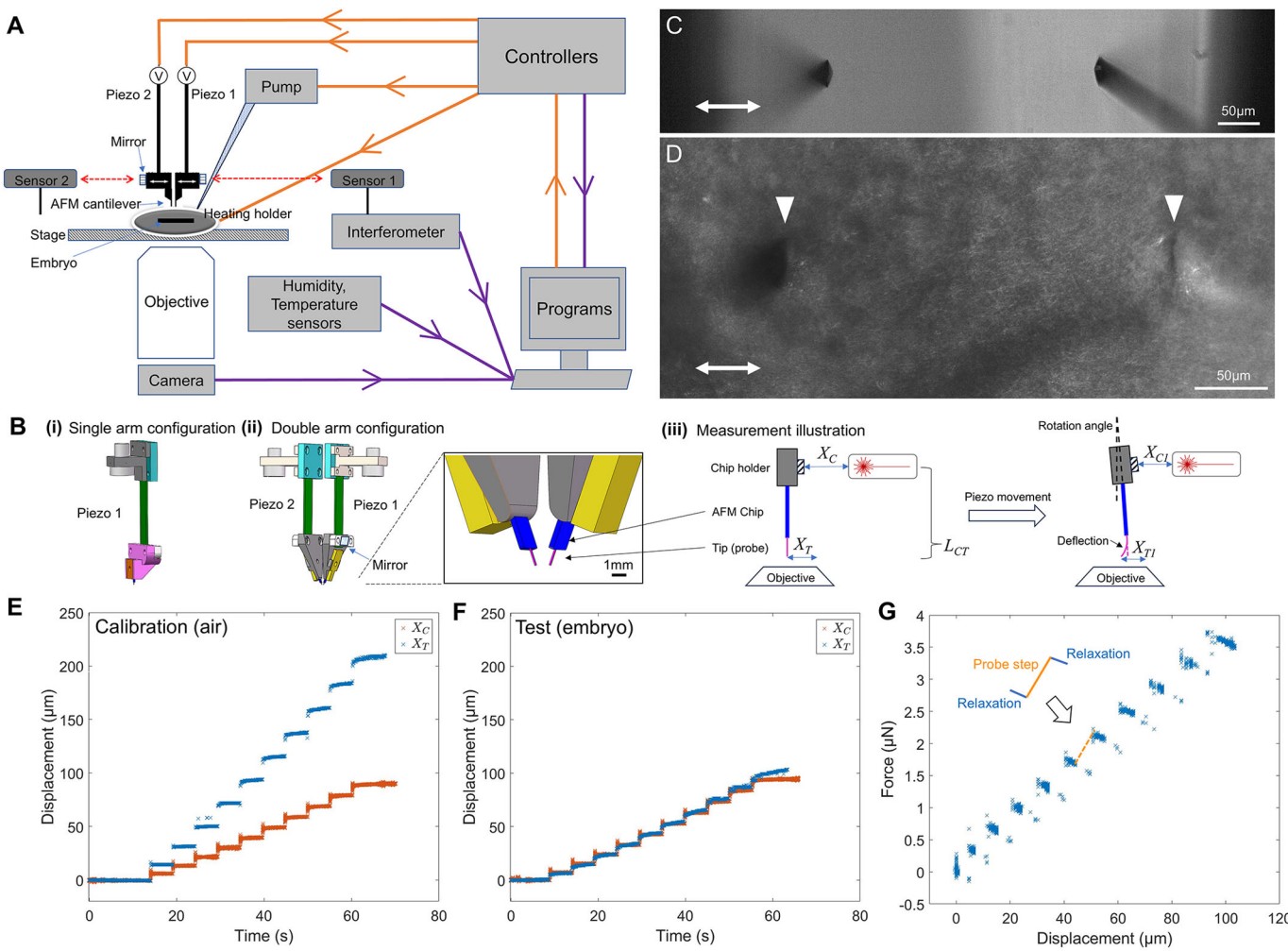

**Fig. 1. TiFM2.0 design.** (A) Schematic of TiFM2.0 highlighting the design logic and main components. (B) Drawing of the single (i) and double (ii) motor probe holder. The blue probe chip and the pink cantilever probes are drawn in exaggerated sizes for illustration purpose. (iii) Illustration of measurement pipeline. $L_{CT}$ marks the distance in $z$-axis between probe tip and the interferometer target mirror. Because of the rotational nature of the piezo, $x_T$ will increase more than $x_C$ under no-load as the system moves in one direction, at a ratio given by $(L_{CT}+L_P)$ $(L_P)^{-1}$, where $L_P$ is the length of the piezo. The rotational angle of the piezo is drawn in exaggerated magnitude for illustration purpose. The $z$-axis movement of the probe tip resulting from this small rotation is negligible. (C) Example of double probes aligned in $z$, microscope view in air (no samples), arrows indicate direction of probe movement. (D) Example of double probes inserted in an embryonic tissue (HH4 epiblast). Arrowheads indicate the inserted double probes. (E) Example of a force loading calibration run in air. The holder was driven to make stepwise movements detected by the interferometer ($x_C$), the images provide tracking of the probe tip ($x_T$). (F) Example of a force loading test run in an embryo (~HH11, medial-to-lateral loading on the body axis), following the control in panel E. Reduced probe movements at each step compared to the trace in panel E indicate sample resistance. (G) Force estimation by comparing data in panels E and F. Stress relaxation is detected by the creep movement of the probe after each step change as a gradual reduction of measured force. See Fig. S1A-F for additional examples.

the tissue to the inside of the probe tips, indicating tension build-up around the edges of the probes during stretching. In contrast, in the pPSM no tearing appeared (Fig. 2A″,A″″). At both locations, an increase of neural tube wall curvature towards the PSM under stretching was observed, indicating close links between the tissues that allow transmission of forces from the transverse contraction of the PSM under stretching to the neural tube. These tissue behaviours are unchanged by the presence of the endoderm, which the probes go through from the ventral side to reach the PSM. We measured the normal strain dynamics of the tissue between the probes effectively as the distance increased between the two probes over the initial distance (Fig. 2B). After each step movement of the motors, the probes in the pPSM were found to stabilise positions more quickly compared to the probes in the aPSM, which showed a more gradual creep. This difference is particularly apparent at higher strains (Fig. 2B, arrows). Upon probe retrieval after 5 min of holding, the

tissues (now tracked manually by the insertion wound sites) both started to recover from the deformation, with the aPSM shortening faster and to a larger extent over time (Fig. 2C). To quantitatively compare the mechanical differences between aPSM and pPSM, we fitted the deformation recovery phase across multiple embryos after probe retrieval using a two-term exponential model which captured the fast (1-5 s) and the slow (20-50 s) timescales (Fig. 2D). The model contained a residual strain term as a result of tissue plasticity. We found variability of maximum and residual strains across embryos, but they were consistently higher for the pPSM than the aPSM in paired measurements within one embryo (Fig. 2E,F). The slow relaxation timescale was smaller for the pPSM, suggesting a more compliant tissue (Fig. 2G,H). These simple fittings provide useful quantifications on the differences between tissue locations, but also show that simple viscoelastic models are insufficient to capture all aspects of the tissue behaviour observed (such as

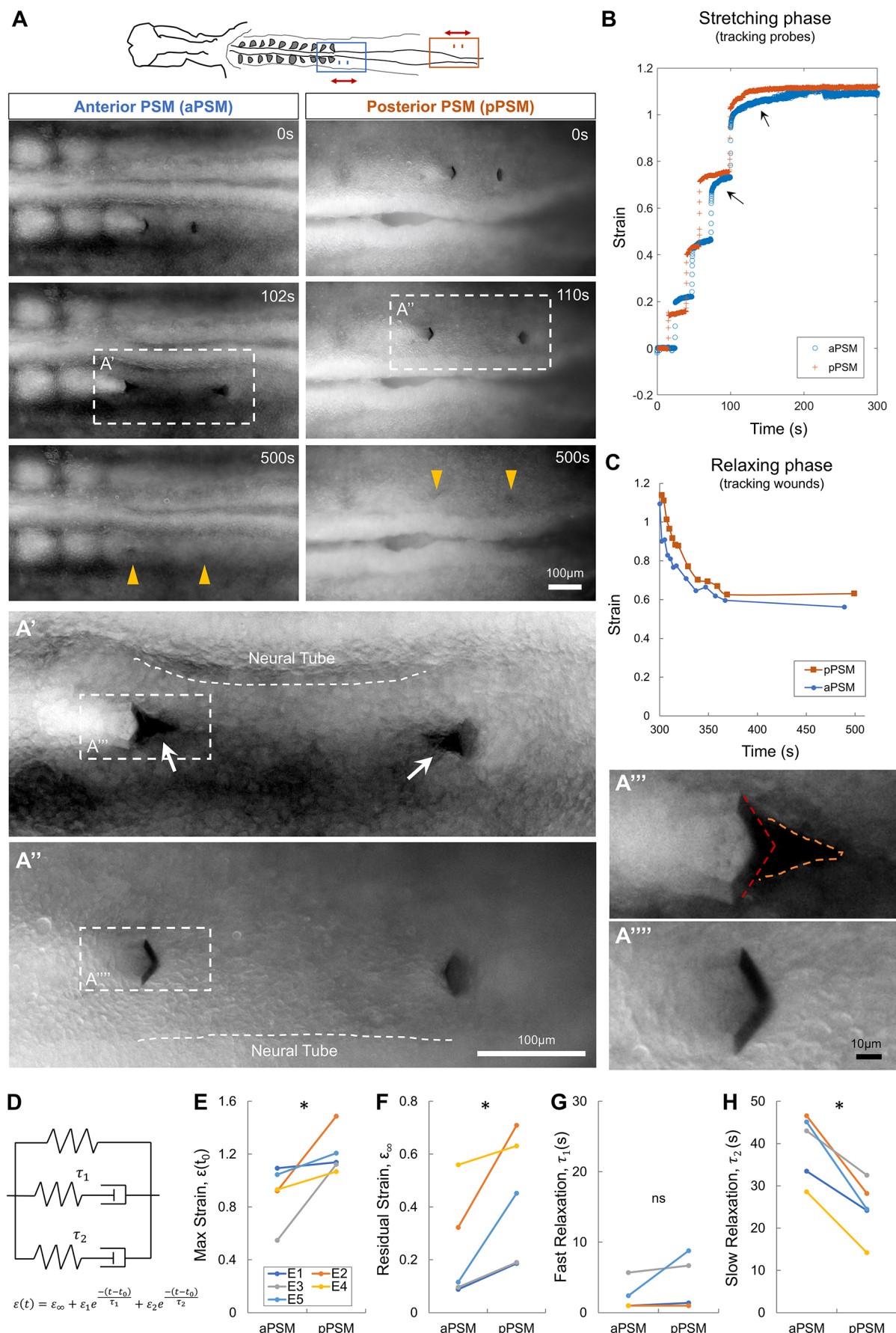

**Fig. 2.** See next page for legend.

**Fig. 2. Local double probe stretching and tissue responses.**
(A) Stepwise stretching along the anterior to posterior axis with two probes ($k$=0.2 N m$^{-1}$; representative of $N$=5 different embryos, ~HH11). The schematic marks the locations in the embryo tested, arrows mark direction of probe movement. Dashed line boxes correspond to zoomed-in views in panels A′ and A″. Yellow arrowheads track the probe wounds after probe retrieval. Dashed lines in A′ and A″ mark the lateral side of the neural tube. Arrows in A′ indicate the tissue tears. A‴ and A⁗ correspond to zoomed-in views showing tissue tearing in anterior and not tearing in posterior. Dashed red lines mark the probe outline, dashed orange lines mark the tearing site. See also Movie 1. (B) Strain dynamics following stepwise stretching. Tracked from the data in panel A. The tissue length between two probes at each step is compared with the original length. Arrows highlight the different creep rates between aPSM and pPSM. (C) Strain dynamics following probe retraction. The time-lapse movies continue from panel B. The probe wounds were tracked manually and the tissue length between two probes is compared with the original length. (D) Diagram of a standard linear solid model for a two-term exponential function fitted to the relaxing phase of the PSM stretching experiments. A residual strain term ($\varepsilon_{\infty}$) is included to account for the plastic deformation. (E) Initial/holding strain from experiment (comparing five embryos each with an aPSM and pPSM measurement, same for panels F-H). Paired one-tailed $t$-test; *$P$=0.029. (F) Residual strain ($\varepsilon_{\infty}$) from the model fit. Paired one-tailed $t$-test; *$P$=0.022. (G) Fast time scale ($\tau_1$) from the model fit. Paired one-tailed $t$-test; n.s., $P$=0.28. (H) Slow time scale ($\tau_2$) from the model fit. Paired one-tailed $t$-test; *$P$=0.001.

plasticity). Overall, these results are consistent with the expectation from recent *in vitro* measurement (Michaut et al., 2025a) that the aPSM is stiffer and more viscous than the pPSM in their native locations *in vivo*. Additionally, the deformation of tissues under the imposed forces can be analysed in terms of cell shape and arrangement changes (Fig. 2A′-A⁗). Cells can be observed to elongate along the expected tension patterns (e.g. extensive between the probes and compressive on the outer sides of the probes).

To further illustrate the potential of combining TiFM2.0 with cellular resolution live imaging, we used the epiblast of HH4 Tg(CAG:memGFP) (Rozbicki et al., 2015) embryos, which allows single cells to be distinguished under the wide-field fluorescent microscope at 200× magnification (Fig. 3A; Movie 2). We performed gradual bi-directional stretching and holding of the epiblast with live imaging, followed by segmentation of the apical surface of epiblast cells using the membrane signal with Cellpose (Stringer et al., 2021) and manual correction. Cells located in the central region between the probes showed an increase in the size and aspect ratio distribution compared to cells in the nearby off-centre region (Fig. 3A-F), consistent with the region between two probes experiencing maximum tension. To examine single cell dynamics during stretching, we tracked individual cells over time. All tracked cells showed an increase in apical surface area (Fig. 3G; Fig. S1G,J; $n$=10/10, 8/8 and 5/5, respectively). The aspect ratio changes of individual cells were varied (Fig. 3H; Fig. S1H,K), as they depend on the angle between the initial orientation of the cell and the stretching direction. This angle indeed decreased in most cells (Fig. 3I; Fig. S1I,L; $n$=9/10, 8/8, 4/5, respectively) upon stretching, indicating cell orientation alignment toward the direction of tension. These results show that TiFM2.0 enables cellular dynamics analysis under mechanical perturbations of the tissue.

To test the capacity of TiFM2.0 to modulate key morphogenetic events, we sought to promote or undo the folding process of the epithelial neural tube (Fig. 4A). During neurulation, the neural plate bends dorsally towards closure at the midline to form the tube, driven by a combination of tissue-intrinsic and -extrinsic forces (Moon and Xiong, 2021), such as apical constriction (Nishimura et al., 2012), apoptosis (Roellig et al., 2022), ectoderm tension (Alvarez and Schoenwolf, 1992; Kunz et al., 2023) and PSM

compression (Xiong et al., 2020). The probes are first calibrated through synchronised control in both compressive and expansive directions (Fig. 4B,C). After anchoring the probes on both sides of the neural fold edge, a sequence was performed to push the neural folds closer together and then to split them (Fig. 4D; Movie 3), using the same conditions as the control sequence (Fig. 4C). Comparing the probe positions in the tissue with the expected no-resistance positions from the calibration curve yields the forces at different folding states (Fig. 4E). The measured forces on the probes are at the 1-10 μN scale between overly-apposed to widely-open neural folds, providing information of the resistance to deformation of the neural plate tissue in its native environment in the folding and unfolding directions (Fig. 4F). These forces are one to two orders of magnitude higher than recently measured native forces that drive folding, such as the compressive force from the pPSM (~100 nN) measured by TiFM1.0 (Chan et al., 2023) and the net compressive forces inside the anterior neural folds (~100 nN) measured by an iMeSH cylinder (Maniou et al., 2024), which drive comparatively much slower tissue deformation (order of 1 h as compared to ~10 s here) during normal morphogenesis of the neural tube.

Using TiFM2.0 we can analyse the heterogeneity of tissue mechanical properties spatially, for example by pulling or compressing the posterior body axis and following its subsequent relaxation (Lu et al., 2024). When driving the probe from posterior to anterior (A-P), we observed higher levels of tissue compression occurring near the probe (close), whilst tissues further anterior (far) showed a decreased amount of deformation (Fig. 5A,E). This is in contrast to driving the probe laterally from left to right (Lateral), where both close (the probe side) and far (the contralateral side) tissue regions showed similar and smaller compression (Fig. 5B,E). Upon probe retrieval after a 5-min holding period, the posteriorly compressed tissues close to the probe partly recovered but retained a significant amount of deformation, in contrast to the tissue far from the probe, which recovered a larger fraction (Fig. 5C-F). In comparison, in the lateral compression, both close and far tissue regions showed recovery (Fig. 5C-F). Despite the difference of the extent of recovery, the timescales were similar (Fig. 5G). These results show that the developing body axis is deformable along the anterior-posterior axis while resistant along the medial-lateral axis. This pattern favours tissue elongation in a bilaterally symmetrical manner. To test this mechanical pattern further in a developmentally relevant timescale, we performed the lateral bending on a cultured embryo using a notochord-anchored probe and held the deformation for 1 h (Fig. 5H; Movie 4). Unlike the 1-min control that elastically restored most of the initial tissue shape, the 1-h embryos exhibited a slow creep (several μm over the hour) during holding and showed significant residual deformations (~50 μm) after probe retraction (Fig. 5H,I). This is far less than the extra elongation along the anterior-posterior axis achievable by a 1-h pulling (~150 μm) under a similar force in our previous study (Chan et al., 2023). These findings suggest that the body axis is less deformable along the medial-lateral direction at longer, developmentally relevant timescales as well.

## Dynamic mechanical measurements
In addition to imposing forces and deformations, TiFM2.0 also enables a variety of dynamic mechanical measurements. For tissue forces, in addition to previously reported single-direction stalling stress measurement (Chan et al., 2023), the second parallel probe allows pinning and clamping of tissues that are not normally held down by other structures. In the example of the early blood vessels, which lay elevated on the ventral side of the embryo above the yolk

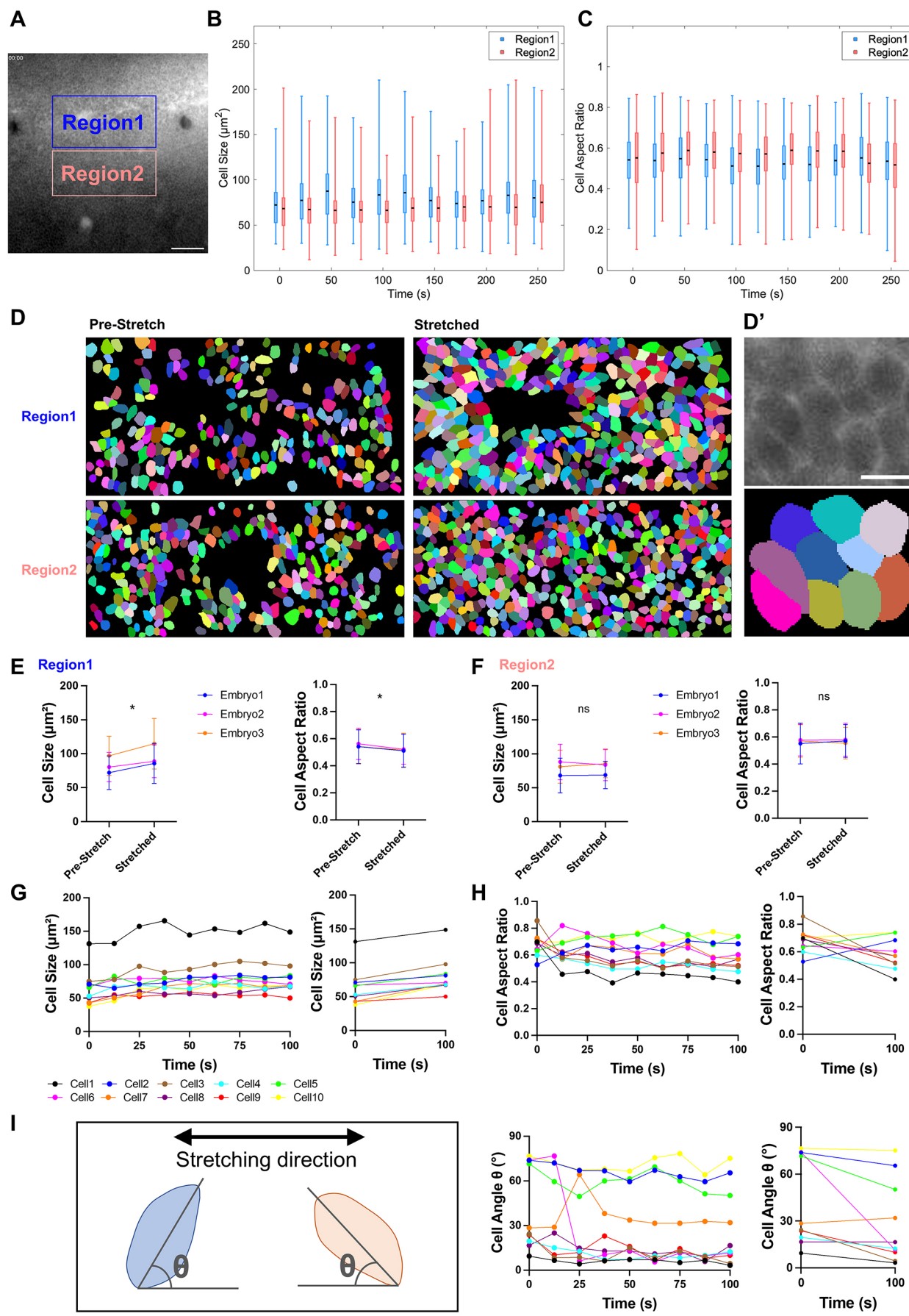

**Fig. 3.** See next page for legend.

**Fig. 3. Cell geometrical changes in response to double probe stretching.** (A) HH4 mem-GFP$^+$ anterior neural plate area was imaged through bidirectional stretching. Two regions were selected for further analysis of cell shape changes. Region 1: located on the central line between the two probes. Region 2: positioned away from central line next to Region 1. See also Movie 2. (B,C) Box plots of cell size changes in the two regions indicated in A over time. T=0 indicates the timepoint before stretching. Within each box, black line marks mean, box extends from the 25th to 75th percentiles of data, whiskers indicate the minimum and maximum values of data. (D) Representative example of cell segmentation masks in the two regions indicated in A before and after stretching. An example of zoomed-in original image and its corresponding cell segmentation mask are shown in D′. (E) Comparison of cell shape changes before and after stretching in Region 1. Left, average cell size changes in each embryo, mean±s.d., *P=0.0378 (n=3). Right, average cell aspect ratio changes, mean±s.d., *P=0.0205 (n=3). Two-tailed paired t-tests. (F) Same as panel E for Region 2. Left, mean±s.d., n.s. P=0.9319 (n=3). Right, mean ±s.d., n.s. P=0.8527 (n=3). Two-tailed paired t-tests. (G) Cell size changes after stretching for ten randomly selected individual cells in Region 1 of a single embryo. Left, cell size changes over time after stretching. Right, comparison of cell size changes before and after stretching. (H) Cell aspect ratio changes for the same cells in G. (I) Cell angle changes for the same cells in G. The schematic defines cell angle (θ) as the angle between the cell major axis and stretching direction. Scale bars: 100 µm (A); 10 µm (D′). See also Fig. S1G-L.

sac, the probes were used to clamp the vessel in place and measure blood pressure (Fig. 6A-D). The heartbeats were unaffected by the presence or engagement of the probes (Fig. 6E). The rhythmic change of blood pressure was absorbed by the expansion of the vessel and was not detectable while the probes were not clamping onto the vessel (Fig. 6C,F). During clamping, the probe tip was observed to move with the vessel wall. The peak-valley difference varied between 5-10 µm, which provided an estimate of pressure differential at 23±4 Pa (n=3, probe force constant k=0.03 N m$^{-1}$, and the contact area A=10$^{-8}$ m$^2$).

TiFM2.0 also enables *in situ* probing of tissue mechanical properties, taking advantage of accurate dynamic positioning and sensitive soft probes. The concept is that a small local motion of the probe tip inside the tissue is locally resisted and also its propagation is gradually dissipated from the source (Fig. 7A). The dynamics of these deformation signals encode the viscoelastic properties of the tissue location. To test this capacity, we inserted the probes into water and a series of poly-isobutylene in hexadecane solutions (PIB, a reference viscoelastic fluid) of increasing storage modulus (Fig. S2A-C). The sending probe was then driven to oscillate using a sine function with a given maximum voltage and frequency while the tip was tracked (Fig. 7A-C). Measuring the amplitudes of $x_T$ in both sending and receiving probes, we resolved differences in the range of 300-3000 Pa in storage modulus of the PIB (Fig. 7Bii), a key range that spans the expected values of embryonic tissues as measured by methods introducing larger deformations (Marrese et al., 2020). At lower frequencies, we increased the measurement sensitivity to the 100 Pa range; however, this sensitivity was lost at higher frequencies (Fig. 7Cii). To further assess the information in the mechanical signal, we fitted $x_T$ to a sinusoidal function and aligned the interferometer signals to the same time axis, allowing phase differences of probe movement between samples to be observed. For example, the probe showed a smaller amplitude and more phase shift in 18.5% PIB compared to water (Fig. S2D), indicating stronger viscoelastic resistance to probe movement in PIB. To test this in epithelial-tissue-like solids, we compared the amplitude of oscillation between PBS and a series of hydrogels of increasing stiffness at two different depths that are representative of experimental conditions in embryos (Fig. 7D; Fig. S2B; Movie 5).

Overall, we observed lower amplitudes as stiffness increased in a quantitative relationship that would allow differences in amplitudes to be related back to differences in the material properties of the samples. Probes with lower force constants (e.g. k=0.03 N m$^{-1}$) can be used for samples at the softer ranges. Of note, at deeper insertions in stiffer gels, the oscillation of the probe tip was antiphase to the imposed sinewave motor motion and was represented as negative, due to the strong resistance bending the probe at a higher z-axis point. Together, these data show that TiFM2.0 can sensitively resolve the mechanical differences of viscoelastic materials in comparable ranges of the embryonic tissues. However, it is important to note that embryonic tissues in their native environment are small and anisotropic, connecting with other tissues through ECM links. The mechanical property and $x_T$ relationships established in the homogenous control materials above cannot be directly used as calibration to estimate values at the tissue location. Extensive mapping using TiFM2.0 in conjunction with proper multi-tissue models are required to quantitatively solve tissue mechanical properties of intact embryos (further discussed in Discussion section).

Nonetheless, TiFM2.0 is readily available to assess developmentally-important mechanical property patterns *in vivo* and *in situ*. Using the oscillatory input as described above, we tested distinct chicken embryo body axis tissues following the parameters of Fig. 7D. The notochord showed a strong restriction on probe movement, whereas PSM was much less resistant (Fig. 7E,F; Movie 6). In the dorsal neural tube, the hindbrain showed less resistance to probe movement compared with the spinal cord (Fig. 7G). These results are consistent with previous studies assessing the deformability of these tissues using other methods (Marrese et al., 2019; McLaren et al., 2025). However, it should be noted that keeping track of the spatial range of tissue deformation by the probe is important for the interpretation of these results. For solid tissues, the resistance to probe movement measured could indicate the neighbouring tissue connections and boundaries rather than local properties around the probe, which can be much stronger. Using the second probe helps to provide a defined spatial range. We tested this in the PSM where small mechanical differences between the anterior and posterior exist, as shown by stretching experiments (Fig. 2B-H). Indeed, the mechanical wave propagated with a smaller magnitude loss in the aPSM compared with the pPSM, indicating that aPSM is more solid-like while pPSM is more liquid-like relatively (Fig. 7H,I; Movie 7). These results are consistent with the reported cell and ECM organisation of the PSM along the anterior-to-posterior axis (Bénazéraf et al., 2010; Michaut et al., 2025a), and recent magnetic-droplet-based rheological measurements in the zebrafish PSM (Mongera et al., 2018), showing conservation of the PSM mechanical property gradient among vertebrates despite distinct tissue sizes and ECM organisations. Assuming local homogeneity similar to control materials near the probe, we estimated the notochord and anterior neural tube stiffness to be in the 1-3 kPa range, whereas the pPSM was in the 0.1-0.4 kPa range. At 0.5 Hz, the storage modulus of the PSM tissue was ∼300 Pa, and shows a small but consistent difference between the aPSM and pPSM. Together, our results demonstrate the capacity of TiFM2.0 for direct *in vivo*, *in situ* tissue mechanical property assessment.

To expand the capacity of TiFM2.0 beyond the thin, flat avian early embryo tissues, we extended the back end of the probe with a DiI bead (Fig. 8A), which provides high-contrast fluorescent signal and additional spatial coverage when the probe front becomes obscured by thicker or less transparent tissues. Using zebrafish blastula-stage embryos, which have a spherical geometry with the yolk and consequently form a thick tissue to image from the side, we

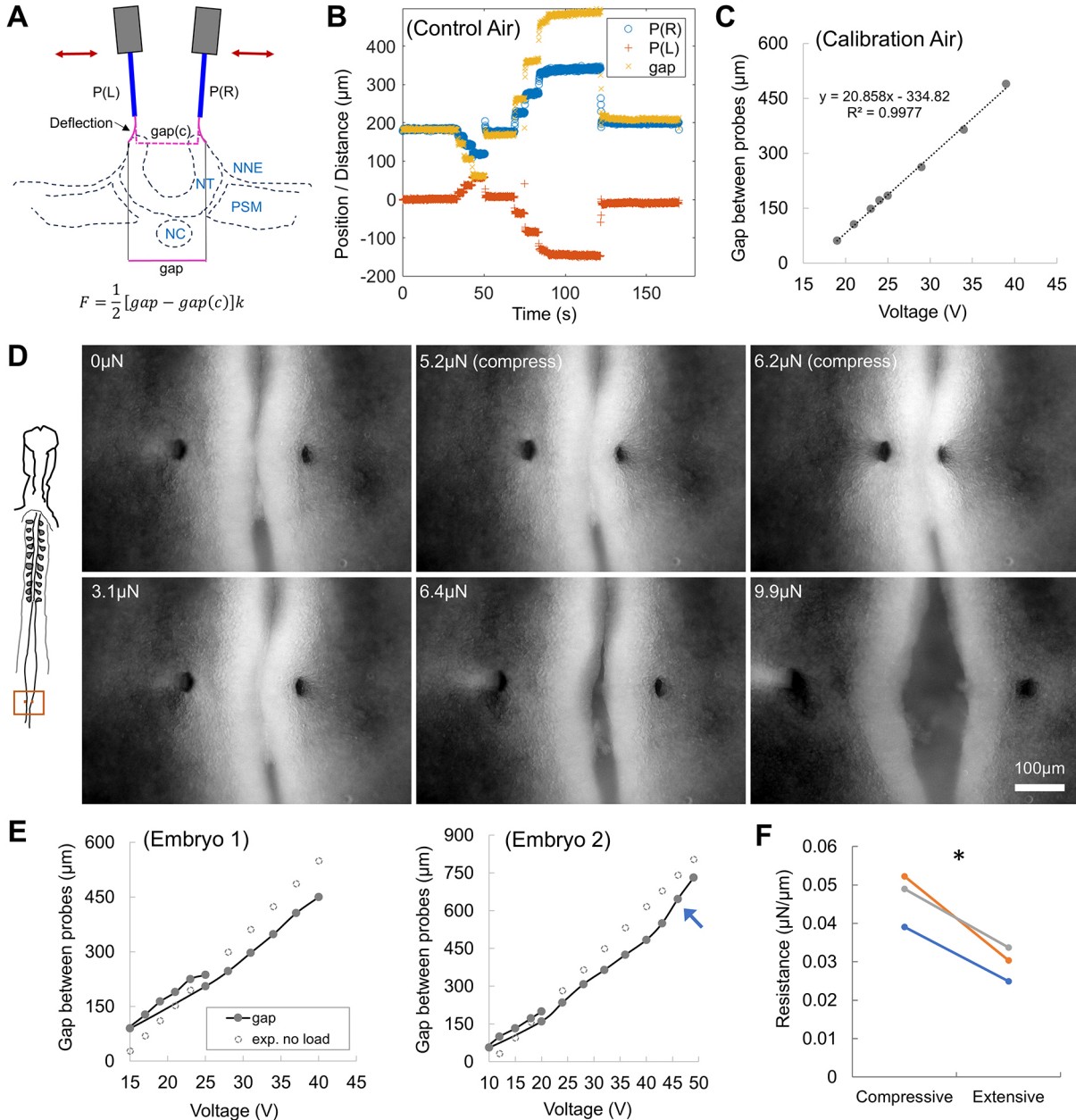

**Fig. 4. Compression and pulling of the neural folds.** (A) Diagram representing experimental design. P(L) and P(R) are the synchronised left and right probes. Gap(c) is the gap between two probes in control/no-resistance condition, compared to the experimental Gap for force measurement. NC, notochord; NNE, non-neural ectoderm; NT, neural tube; PSM, presomitic mesoderm. (B) Double probe tracks from a control time-lapse movie. (C) Calibration of probe gaps with the driving programme to establish the no-force gap curve. (D) Images of the compressed and pulled neural folds under the driving programme. Representative sample from $N$=4 embryos, ~HH11. The parallel bright tissues in the middle are the two neural folds. Forces labelled (in µN) represent force exerted on the probes ($k$=0.2 N m$^{-1}$) by the neural folds during compression and pulling. See also Movie 3. (E) Probe gaps over the course of the perturbation for two representative embryos. Black line links the steps, starting from 25 V. A roughly linear resistance increase is seen in both folding and unfolding directions. Arrow marks tissue yielding when the force exceeds the elastic regime. (F) Tissue resistance fitted from the force-displacement relationship. The neural tube shows more per unit displacement resistance in the folding direction compared with the unfolding direction. Paired two-tailed $t$-test, *$P$=0.019.

tapped the trailing probe on the animal pole tissue, using fish medium and a later somitogenesis-stage embryo as controls (Fig. 8B,C; Movie 8). Shortly after probe contact with the animal pole, the front became obscured by the tissue and could no longer be tracked, but the extended dye was clearly trackable (Fig. 8C). This allowed an accurate determination of the animal pole blastoderm deformation to the probe (Fig. 8D-F), which suggests that it is softer than the dorsal tissue of the later-stage embryo, as evidenced by the consistently larger probe displacements observed at each loaded

voltage (10 V, 30 V and 50 V). These experiments show the feasibility of TiFM2.0 for side measurements of thicker tissues and a different model organism following probe modifications.

## DISCUSSION

In this work we provided proof-of-concept demonstrations of the versatility of TiFM2.0 in probing and perturbing tissue mechanics in early vertebrate embryos, with precision, flexibility and tissue-depth coverage at biologically relevant timescales and forces. These

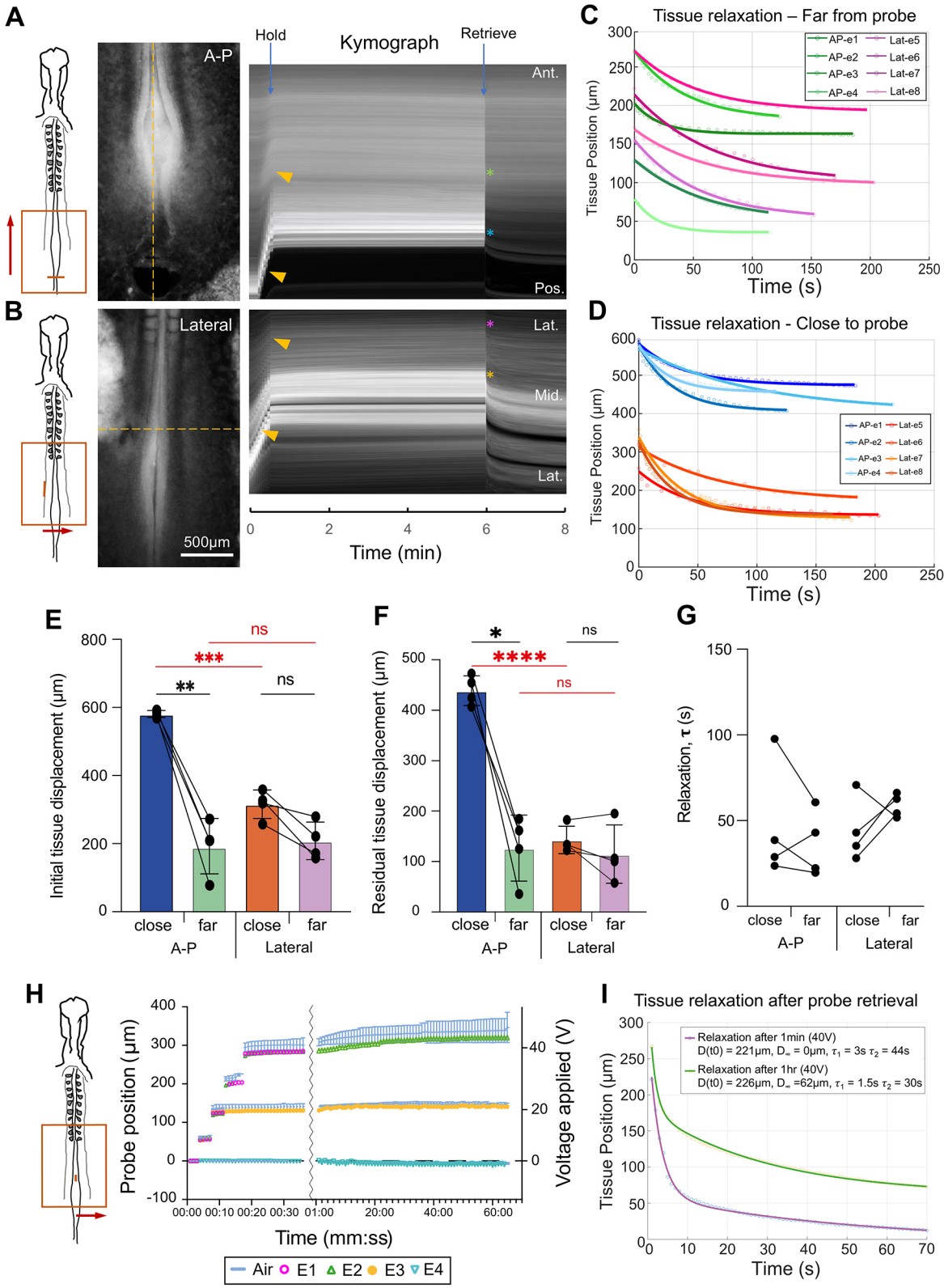

**Fig. 5.** See next page for legend.

experiments can reveal new biological insights, in combination with other mechanical tools, imaging and molecular approaches (Lu et al., 2024; McLaren et al., 2025). Notably, the collection of evidence so far starts to constitute a new level of understanding on the tissue mechanics of the avian embryonic posterior body. The

tissues are produced at the posterior end and follow a posterior-to-anterior differentiation gradient that is now also revealed to be in parallel with a mechanical property gradient. The PSM cells, upon exiting the progenitor domain and entering the PSM, simultaneously drive tissue expansion and softening as they exhibit

**Fig. 5. Mechanical anisotropy of anteroposterior and mediolateral axes.**
(A,B) Posterior body axis anteroposterior (A) and mediolateral
(B) compression using an aluminium foil probe (~400 µm wide) on ~HH10
embryos. The probe was inserted just posterior to the neural tube and
moved in the anterior direction ~600 µm (A) or inserted at the PSM-lateral
plate mesoderm (LPM) border and moved ~400 µm towards the midline (B).
The yellow lines indicate the axes used for kymograph plotting (shown on
the right). The kymograph is labelled with the points at which the maximum
compression was achieved (Hold) and when the probe was retrieved
(Retrieve), ~5 min 30 s later. Arrowheads in each image compare the extent
of deformation under the probe at different distances from the probe, where
A shows more drop of slope between bottom and top arrowheads compared
to B. Asterisks compare the extent of recoil close and far from the probe.
(C,D) Position of tissues tracked following probe retrieval after holding
compression in anteroposterior (A-P) and lateral (Lat) directions. One term
exponential decay with a residual term (line) was fitted to the deformation
tracks (dots). Tracks of tissue far (C) and close (D) to probe are compared.
(E-G) Maximum tissue displacement before retrieval (E), fitted residual tissue
displacement (F) and relaxation timescale ($\tau$) after compression and hold for
~5 min (G) in A-P and lateral compressions, close and far from the probe.
Mean±s.d. In black, two-tailed paired ANOVA; **$P$=0.0069 (E) *$P$=0.0137 (F).
In red, two-tailed Welch's $t$-test; ***$P$=0.0007 (E) ****$P$≤0.0001 ($n$=4) (F).
ns, not significant. (H) Mediolateral compression using a notochord anchored
probe (~50 µm wide) in ~HH11 embryos. Air controls show normal probe
movement ($n$=20, mean±s.d). E1 shows compression for ~1 min. E2 shows
40 V of compression for 1 h. E3 shows 20 V compression for 1 h. E4 shows
no compression for 1 h. Embryo resists probe movement. During
compression, images taken at 1 s interval. During holding, images taken at
1 min interval. Also see Movie 4. (I) Tissue relaxation of E1 (after 1 min of
compression) versus E2 (after 1 h of compression). Curves obtained from
fitting a two-term exponential function with a residual displacement to the
tracks. $D$(t0) maximum tissue displacement, $D_\infty$ residual tissue displacement,
fast ($\tau$1) and slow ($\tau_2$) tissue relaxation estimation. The difference in
$D_\infty$ indicates plastic deformation in longer-term.

high motility and reorganise the ECM to absorb water (Bénazéraf
et al., 2010; Lu et al., 2024; Michaut et al., 2025a). This orients PSM
deformation towards the posterior, and promotes the deformation of
axial tissues such as the neural tube towards narrowing (including
convergence and folding) and extension (Chan et al., 2023;
Regev et al., 2022; Xiong et al., 2020). The stiffer axial tissues
push through the 'melting' progenitor domain (undergoing epithelial-
mesenchymal transition; Goto et al., 2017) containing new PSM
cells, bisecting them to supply the bilateral sides (Xiong et al., 2020).
These mechanical patterns and interactions enable directionality
(Bénazéraf et al., 2010), tissue coupling (Xiong et al., 2020) and a
robust speed (Lu et al., 2024), as well as potentially other key features
of the body axis such as bilateral symmetry. TiFM here provides
important direct measurements and controlled perturbations that put
this description into proper physical context.

An important next step is to quantitatively interpret mechanical
patterns and heterogeneities in the embryonic tissues uncovered by
TiFM2.0. The *in situ* probing of tissue mechanics at the scale
relevant for morphogenesis (~30-100 µm) faces the challenge of an
inhomogeneous and complex surrounding that in many cases can
make the source of stress/resistance to the probes unclear, as early
embryonic tissue sizes are comparable to the scale of measurement
(cell sizes at these stages are on the order of 10 µm so the probes
primarily measure tissue-level properties). Physically separating
different tissues (such as by microsurgery) that could contribute to
the probe dynamics would likely disrupt key ECM links between
tissues and introduce mechanical effects at the wounds (such as
tension increase), limiting the effectiveness of experimental dissection
of different contributors. To address these challenges, a multi-tissue
model that allows systematic fitting of TiFM data to the complex
tissue environment is needed. To start, the complex modulus of the

control materials will be recapitulated with the probe geometry
and measurement dynamics. This will allow characterisation of the
strain field around the probe in the materials. Next, the model will
be extended by supplying inhomogeneous samples, such as two
connected but distinct tissues (e.g. PSM and neural tube) where
assumptions are made about their material properties and their
ECM connections. Experimental data, such as mechanical signal
propagation along and across tissues, conducted systematically at
different depths, frequencies, directions and locations will then be
used to fit the parameters of the model. Here, it will be important to
consider the property of the tissues that need to be imaged that defines
the range that the probes are detecting . For example, a solid tissue like
the notochord will propagate the deformation to far distances, whereas
the PSM will dampen it near the actuation site. Still, even the most
fluid-like pPSM propagates a fraction of deformation to far distances,
showing the multilayered structural complexity of tissues that needs to
be taken into consideration. Finally, TiFM actuation will be deployed
to directly test, for example, the strength of ECM connections between
tissues. How well this approach will be able to address the complex
and heterogenous mechanical environment of the tissue remains to be
explored. An experimental model of intermediate complexity may be
beneficial, such as multi-tissue body axis organoids (Hamazaki et al.,
2024) created *in vitro*. These steps will establish a comprehensive
picture on how tissue properties coordinate complex morphogenesis
and realise the full potential of TiFM systems. The avian embryonic
body axis here provides an excellent model for this type of systematic
deconstruction of mechanical interactions. The understanding
gained will illuminate human development during these stages
when many defects arise, and where comparable processes have not
been described due to the scarcity of embryo samples. More generally,
the rationale and approaches of TiFM applied here to the body
axis model are widely relevant to other classes of fundamental
morphogenesis processes.

While TiFM2.0 as described in the current work contains a variety
of modules and a complex user protocol, the core mechanism and
function of TiFM systems as a controlled cantilever are simple (Chan
et al., 2023; Michaut et al., 2025b preprint; Oikonomou et al., 2025)
and can be deployed in a compact, easy-to-construct version (see
Table 1 – simple TiFM column) or upgraded to work with high
resolution microscopy, suited for large tissue-level measurement and
perturbations or dynamic observation of cellular and molecular
changes associated with tissue forces. Moreover, the system can be
combined with other mechanical modules such as tension (Kunz
et al., 2023) and magnetic (Serwane et al., 2017) controllers to enable
multiplexed mechanical measurement and perturbation. Continued
innovations in probe design and positioning as well as theoretical
modelling for data quantification will further improve the
applicability of TiFM systems and reduce their footprint and cost,
enabling potentially wide use in developmental biology research and
other settings.

## MATERIALS AND METHODS
### List of hardware components
An equipment and component list is provided for the TiFM2.0 system in
Table 1. The table is organised by different modules. A simple/minimal TiFM
setup is highlighted by selective components. It is not necessary to procure
identical items. In particular, accessories such as connecting parts and scaffolds
should be chosen according to the configuration of the base microscope. Basic
instrument construction knowledge is needed to assemble the systems.

### Working environment and maintenance
TiFM systems should ideally be set up with vibration-isolation in quiet areas
with stable temperature and humidity, and minimal dust. The TiFM2.0

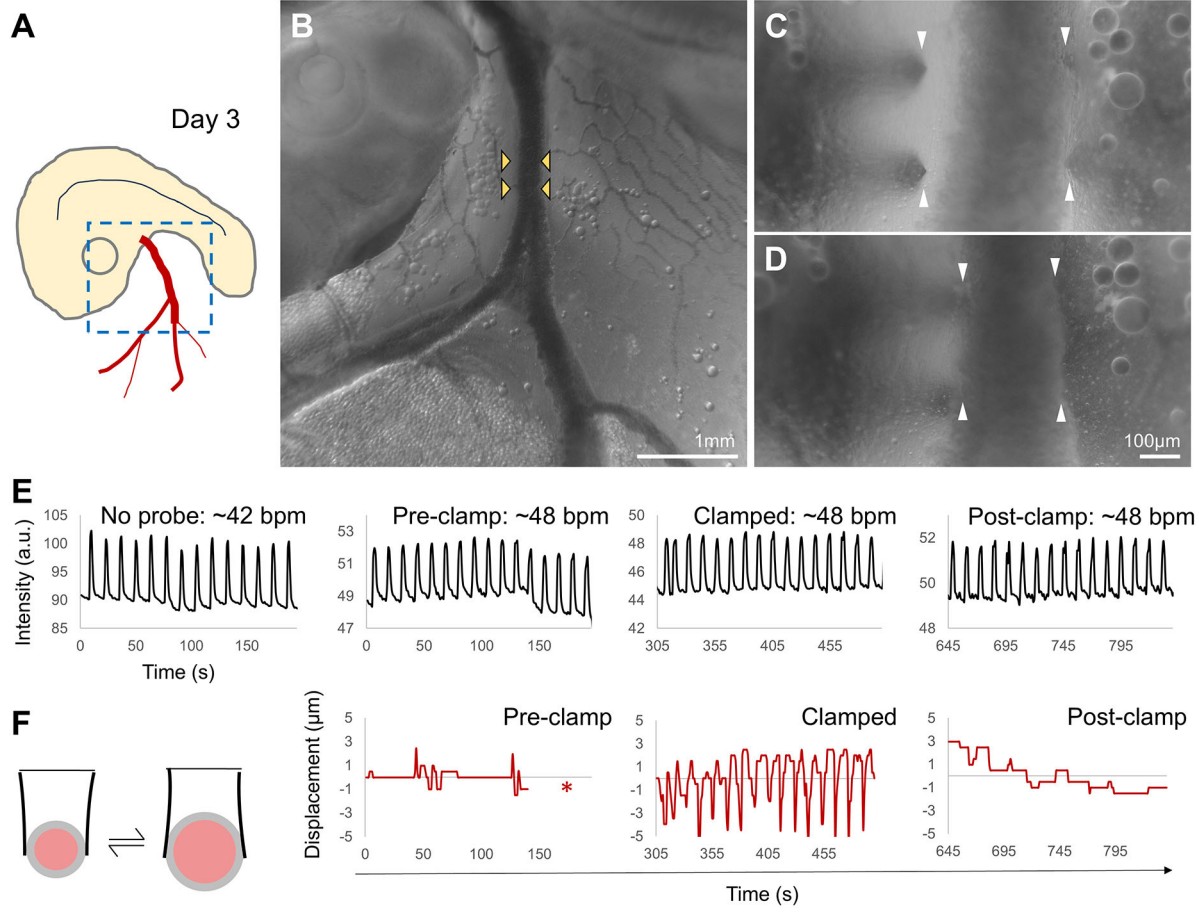

**Fig. 6. Blood pressure in the embryonic artery.** (A) Diagram of a day-3 embryo showing the imaged region and the right omphalomesenteric artery leading off the body. (B) Zoomed out image of the region and the artery. Yellow arrowheads mark the insertion points of the double-pronged probes on both sides of the artery. (C) Probes pre-clamp. The right-side probes are in contact with the artery wall. (D) Probes clamping the artery after the left-side probes are moved to engage. (E) Monitored heart rate of the embryo through the experiment. (F) Diagram and measurement of probe movement. Heartbeats send a surge of blood flow through the artery and expand the vessel, generating a transient pressure on the probes. Asterisk in the Pre-clamp plot marks the time period when the probes are being moved with no displacement measurements.

interferometer positioning system is robust to sample humidity and does not require regular adjustment and cleaning of the capacitor sensors in TiFM1.0, but the environmental temperature and humidity should still be recorded as they can affect the sensor parameters. All wire connections should be checked regularly to ensure proper connection. The optical fibre wires should not be acutely bent or under tension. The piezo, probe holder and probes should be uninstalled from the probe head micromanipulators and stored in safe boxes when not in use as they are fragile. When installing the probe chips, a stand for the probe holder is recommended to free both hands. Under good lighting where the thin cantilever(s) is visible by naked eye, the user picks up the probe chip with a pair of sharp tweezers, holding the chip while staying away from the end with cantilevers, and inserts the chip into the holder slot, before tightening with the screwdriver. When installing the probe holder to the scaffold and micromanipulators, take care to securely connect the piezo cables to the out cable from the voltage controller with the right match to avoid damaging the piezo. Upon finishing the experiments, use a clean imaging dish filled with deionised water to rinse the probe by dipping the chip/probe into the water a few times using the micromanipulator, before uninstalling the probe and holders.

### Operation protocols
When ready to use the TiFM system, power up all hardware and open all controller software on the PC. In the following Ximea Cam tool (Ximea), MT voltage controller (Thorlabs), and Picoscale interferometer software (SmarAct) are used as examples to outline the protocol, which can be adapted to other hardware and software accordingly. First, check if the acquisition systems are functioning normally by starting live imaging, and change voltage input to move the piezo. Next, prepare an embryo sample and mount it on the imaging dish as desired for the actual measurements, add a few drops of PBS/Ringers solution on top to prevent drying (thin albumen is also ok with strong probes such as 0.2 N m$^{-1}$ but may damage the weaker ones due to its adhesion and viscosity). Mount the dish on the stage, position it so that the area of interest is centred and turn on the gooseneck LEDs to provide illumination. Use 2.5×, 5× and 10× objectives to observe the embryo with the camera, adjust the lighting angles and exposure times as necessary. Next, focus on the tissue of interest under 10×, record the focus positions of the tissue surface and the depth of interest, leaving the focus unchanged for probe adjustment. Remove the sample dish and lower the scaffold holding the piezo and chip/probe, carefully approach the stage to avoid collision with the objective and centre the probe tip at the pre-determined focal plane using the XYZ micromanipulator. This ensures the tip and the tissue of interest will be aligned. Use the second YZ micromanipulator to align the second probe to the desired relative position with the first probe, usually at the same focal plane/z-axis position, if applicable.

When the probes are in position, the interferometer sensor should be aligned to the probe holders. First align the sensor head roughly with the holder mirror using the guide laser. Then in the Picoscale software, use the 'Adjustment' mode to obtain the signal pattern on screen, while adjusting the position and angle of the sensor holder to improve the alignment as evaluated by the pattern score, using the micromanipulators of the mirror mount. Leave the sensor holders untouched once a good signal is obtained

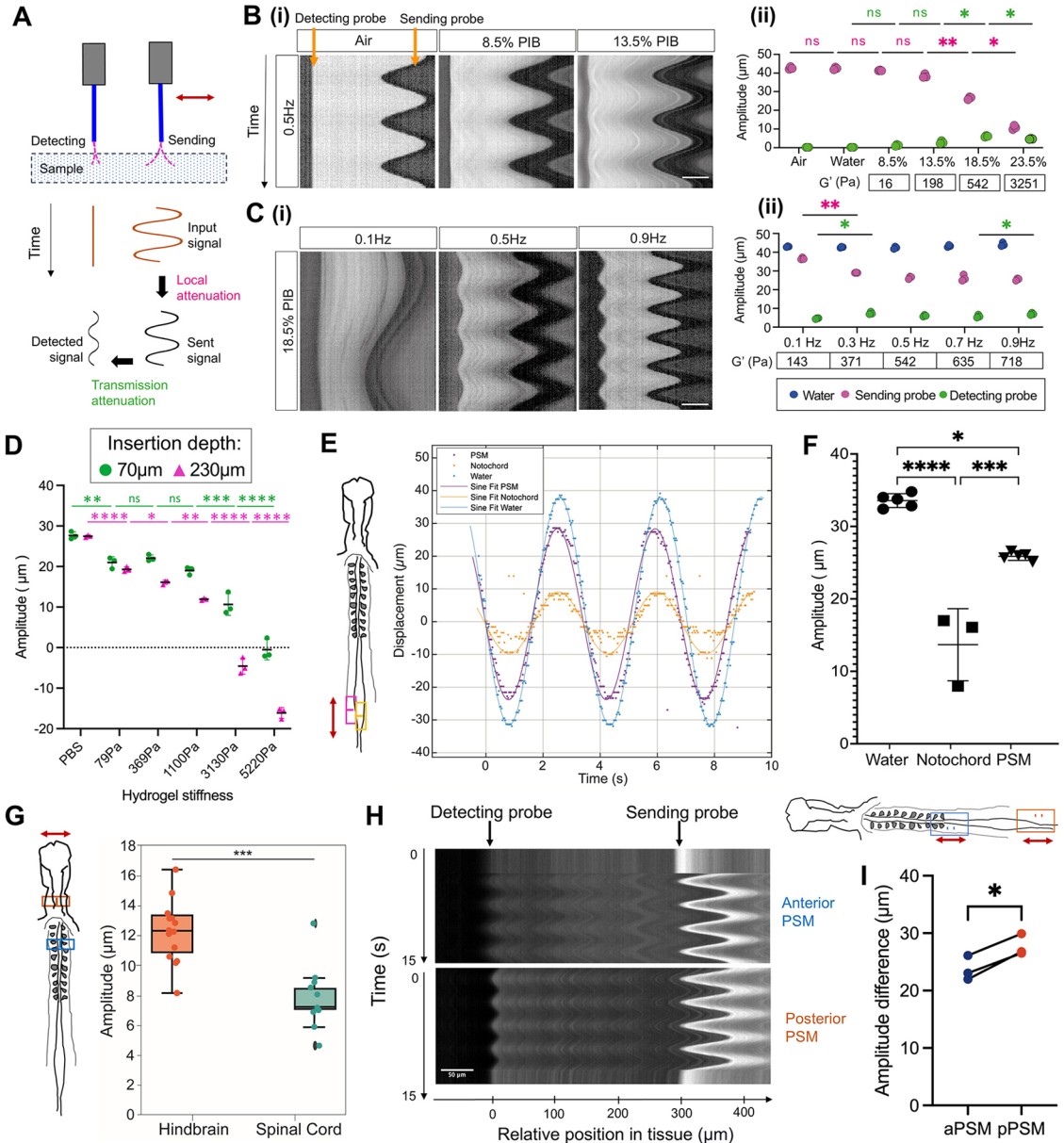

**Fig. 7. Tissue mechanical evaluation with oscillatory probe movements.** (A) Diagram of oscillatory motion used to assess the mechanical properties from the displacement response. The input signal is attenuated by tissue resistance at the origin as well as through transmission. The spatial-temporal dynamics of the signal (e.g. magnitude and phase) encode viscoelastic information of the tissue location. (Bi,Ci) Representative kymographs showing sending and detecting probe movement. Scale bars: 50 µm. (Bii) Amplitude of sending and detecting probes ($k$=0.5 N m$^{-1}$) in different PIB concentrations. Input signal: 10 V, 0.5 Hz sinewave frequency. Probe insertion depth 70 µm (±20 µm). Tukey's multiple comparison test ($n$=3). Sending probe: 13.5% versus 18.5%, **$P$=0.0032; 18.5% versus 23.5%, *$P$=0.0162. Detecting probe: 13.5% versus 18.5%, *$P$=0.0290; 18.5% versus 23.5%, *$P$=0.0379. (Cii) Amplitude of water, sending and detecting probes ($k$=0.5 N m$^{-1}$) in 18.5% PIB at different frequencies. Input signal: 10 V. Tukey's multiple comparison test ($n$=3). Sending probe: 0.1 Hz versus 0.3 Hz, **$P$=0.0087. Detecting probe: 0.1 Hz versus 0.3 Hz, *$P$=0.0442; 0.7 Hz versus 0.9 Hz, *$P$=0.0288. G′ values of PIB samples were measured using a rheometer. See also Fig. S2C. (D) Amplitude of probe ($k$=0.2 N m$^{-1}$) differences for PBS and hydrogels of varying stiffness. Input signal: 10 V, 0.5 Hz. Insertion depths: circle 70 µm and triangle 230 µm. Negative amplitude is defined as tip movement in the opposite direction of the driving motor. Black line marks mean±s.d. Two-tailed Tukey's multiple comparison test ($n$=3). At 70 µm: ns 79 Pa versus 369 Pa, $P$=0.9685; ns 369 Pa versus 1100 Pa, $P$=0.3101; **$P$=0.0051; ***$P$=0.0008; ****$P$<0.0001. At 230 µm: *$P$=0.0258; **$P$=0.0037; ****$P$<0.0001. See also Movie 5. (E) Representative example of probe tip position tracking (marked by dots) and sine fitting (marked by line) across Water, PSM and notochord measurements (HH9-10 embryos). Sine fitting over $n$=2.5 periods. See also Movie 6. (F) Amplitude differences between Water ($n$=5), Notochord ($n$=3) and PSM ($n$=5) in HH9-10 embryos. Probe force constant $k$=0.2 N m$^{-1}$, 10 V, 0.5 Hz. Mean±s.d shown. Tissue insertion depth=70 µm. Two-tailed Tukey's multiple comparison test: *$P$=0.0152, ***$P$=0.0007, ****$P$<0.0001. (G) Amplitude of probe ($k$=0.02 N m$^{-1}$) movement driven at 0.2 Hz and 5 V in the hindbrain ($n$=14) and spinal cord ($n$=10) of HH11-12 chicken embryos. Mann–Whitney $U$-test, ***$P$<0.001. Box plots show median line and the first and third quartiles (box bounds), whiskers show 1.5× interquartile range. (H) Kymographs representing double probe movement tracks for anterior (aPSM) and posterior (pPSM) locations. Representative of $N$=3 samples, HH10 embryos. Sending and detecting probe amplitude tracked by high contrast lines. $k$=0.03 N m$^{-1}$, 10 V, 0.5 Hz. Scale bar: 50 µm. See also Movie 7. (I) Amplitude difference between sending and detecting probes in anterior PSM (aPSM) and posterior PSM (pPSM). A smaller amplitude difference shows the detecting probe oscillates more in response to the sending probe, suggesting the tissue is stiffer. A large amplitude difference indicates the detecting probe does not respond to the imposed oscillation from the sending probe, suggesting a softer tissue. Two-tailed paired $t$-test; *$P$=0.0176, $n$=3.

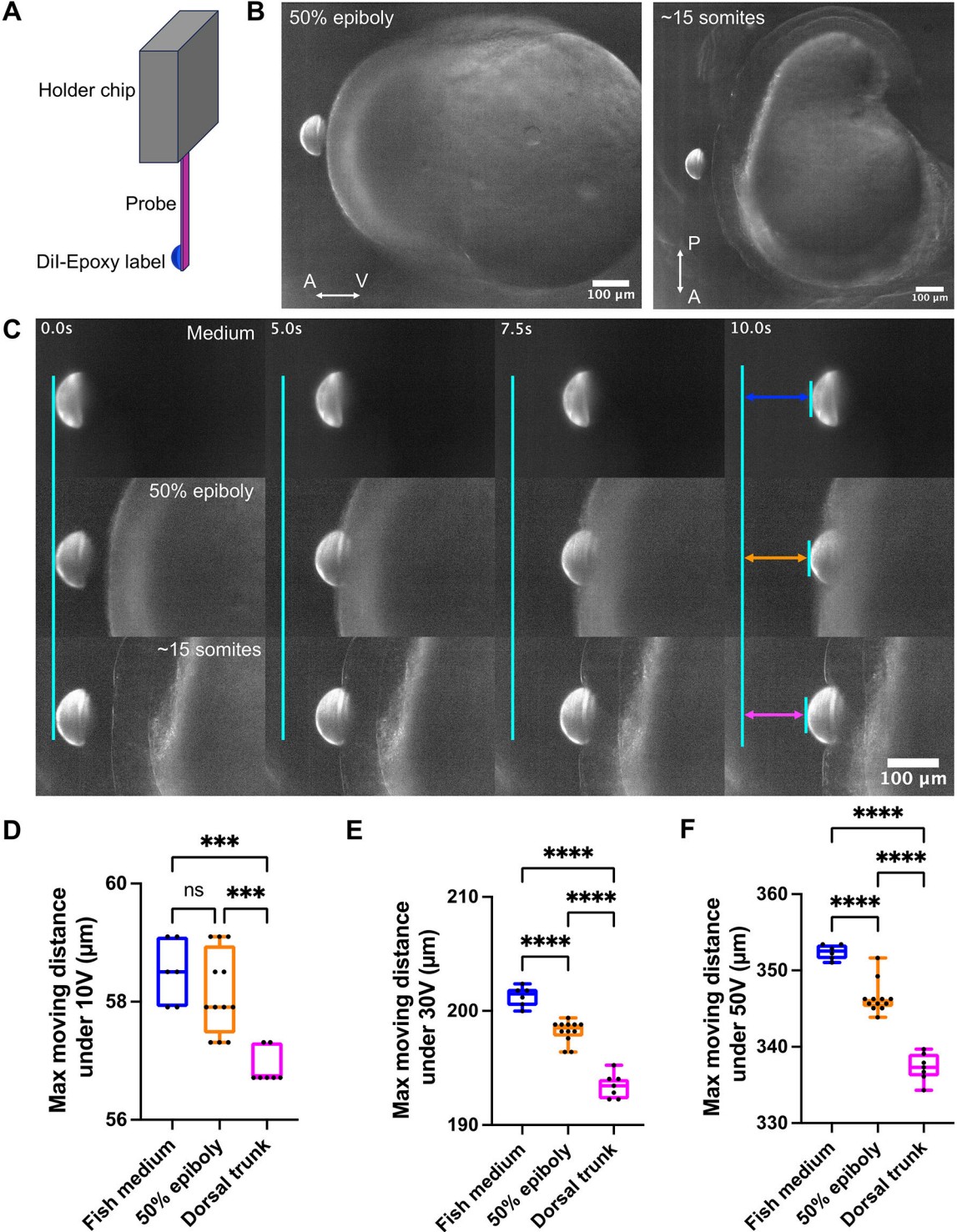

**Fig. 8. Tracking modified probe in thick tissues in zebrafish embryos.** (A) Diagram of the probe with a DiI-Epoxy label. (B) Example images of the labelled probe and the sample embryo. The animal pole at 50% epiboly (left) and the dorsal trunk during somitogenesis (right). Movies are taken with red fluorescence and a low level of background light to visualise the embryo tissues. A-V axis indicates animal-vegetal; P-A axis indicates posterior-anterior. (C) Imaging of the probe movement when pushing different tissues under the voltage load on the probe of 20 V, 0.1 Hz and Ramp function (linear movement of the motor). Fish medium (1st row), animal pole at 50% epiboly (2nd row) and dorsal trunk at ∼15 somite stage (3rd row) are shown here as representative examples. Images show the probe position at the beginning of loading the voltage (1st column), starting to touch the tissue edges (2nd column), continuing to push the tissues (3rd column) and the final position when the maximum voltage is reached (4th column). The blue, orange, and magenta arrows indicate the maximum moving distance of the probe under pushing in fish medium, epiboly and dorsal trunk, respectively. See also Movie 8. (D) Resistance comparison (shown by maximum moving distance of the probe, tracked by the bright edge kymograph of the backend of the probe, same as panel E and F) of fish medium, animal pole at 50% epiboly and dorsal trunk at 10 V maximum displacements. Ordinary one-way ANOVA multiple comparison test. Fish medium ($n$=6) versus 50% epiboly ($n$=12), n.s=0.4581; fish medium ($n$=6) versus dorsal trunk ($n$=7), ***$P$=0.0001; 50% epiboly ($n$=12) versus dorsal trunk ($n$=7), ***$P$=0.0003. (E) Resistance comparison of 30 V maximum displacements in the same embryos shown in D. Ordinary one-way ANOVA multiple comparison test, ****$P$<0.0001. (F) Resistance comparison of 50 V maximum displacements in the same embryos shown in D. Ordinary one-way ANOVA multiple comparison test, ****$P$<0.0001.

**Table 1. An equipment and component list for the TiFM2.0 system**

| Module | Component | Function and notes | Source | Part number or link | Simple TiFM |
|---|---|---|---|---|---|
| Probe head | AFM chip and probes | Inserting into sample as a cantilever | Brukers, NanoWorld, BudgetSensors, Mikromasch | MLCT-O10, ARROW-TL2, AIO-TL, TL-CONT, HQ:CS38 | x |
| | Electrical piezo | Controlling chip location in the *x*-axis | Thorlabs | PB4NB2W | x |
| | Custom chip holder | Fitted with reflective mirrors | This study* | | x |
| | Custom piezo holder | – | This study* | | x |
| | Various connecting rods | Selected depending on the configuration | Thorlabs | https://www.thorlabs.com | x |
| | (Master) micromanipulator XYZ | Adjusting the overall location of probe 1 and 2 relative to the sample | World Precision Instruments | WPI M3301R | x |
| | (Secondary) micromanipulator YZ | Adjusting the location of probe 2 for alignment | Thorlabs | NF15AP25 | – |
| | Rails and support scan folds | Supporting the probe head construct and connecting to the base microscope | Thorlabs | https://www.thorlabs.com | x |
| | Wires and connectors | Connecting the piezo to the voltage controller | Rapid Electronics | https://www.rapidonline.com/ | x |
| | Voltage controller | Sending driving programmes to the electrical piezo, voltages are related to $X_C$ although offsets and an error margin exist | Thorlabs | MDT693B | x |
| Base microscope | Inverted widefield and fluorescence. Axio Observer 7, 2.5/5/10/20× objectives, *xy* motorised stage, side camera port, fluorescence filters and LED light source for RL imaging, objective holder *z*-axis recording, associated controllers | Visualisation of the probes ($X_T$) and the sample. Different magnifications are essential for adjusting probe location followed by fine alignment. *Z*-axis recording is essential for determining probe insertion depth. Top illumination and condenser removed to allow installation of the probe head | Zeiss | https://www.micro-shop.zeiss.com/en/uk/system/inverted+microscopes-axio+observer-microscopes/6001/ | x (any inverted scope) |
| | Goose neck top illumination | Providing side illumination and flexibly enhancing contrast of probe images | VWR | 631-1086 | – |
| | Camera | Depending on applications needed, such as spatial temporal resolution, camera speed and sensitivity are chosen accordingly | Ximea | MQ042RG-CM | x (any camera) |
| Interferometer | Picoscale interferometer and breakout box | Providing precision measurement of the location of the probe holder ($X_C$) | SmarAct | PS-CTRL-V2.0, PS-BOB-V2.0 | – |
| | Sensor head F01, F04 | Working distance and angular tolerance need to be selected to balance the alignment requirement while keeping proper distance from sample and probe holders | SmarAct | PS-SH-F01, PS-SH-F04 | – |
| | Optical fibre cables | | SmarAct | PS-ACC-PAT-APC-5M-APC-D | – |
| | Scaffolds and rails | Selected depending on the configuration | Thorlabs | https://www.thorlabs.com | – |
| | Micromanipulator XYZ | Adjusting interferometer sensor head in XYZ | Thorlabs | XR50P | – |
| | Adjustable sensor holder | Adjusting angle of interferometer sensor heads | Thorlabs | MK05 | – |
| | Trigger signal generator, Arduino linking camera and breakout box | Enabling synchronisation of the camera (tip image, $X_T$) and the interferometer sensor ($X_C$) for real time stress dynamics | Rapid Electronics | 75-0999 | – |
| Incubation | Programmable fluid pump | Providing a steady and slow water flow to maintain sample humidity | World Precision Instruments | AL-4000 | – |
| | Heating insert with sensors and feedback control | Maintaining stable temperature for the sample | Warner Instruments | TC-324C, DH-35 | – |
| | Fluid tubing and glass pipette | Guiding water to the required sample area | n/a | – | – |
| System computer | PC with >10 USB ports | Controlling all components and recording data | Dell | https://www.dell.com/ | x |

*Custom components include the motor and AFM chip holder module, and were printed by a 3D Printer (UltiMaker 3, Print Core AA 0.40) using PLA filament (2.85 mm, RS PRO 832-0273). A tailored mirror (Silicon wafer, N-Phos) was glued to the outer side of the AFM chip holder for interferometer positioning. The engineering models and other details are available upon request. Custom probes were made by glueing aluminium foil (wide probe used for compression) or DiI-Epoxy beads to the end of the AFM cantilevers.

**Table 2. Practices for minimising technical variability**

| Source of error | Context of error | Magnitude | Mitigation |
|---|---|---|---|
| **Probe positioning** | | | |
| Manual probe tip identification and tracking | Probe location is identified as the pixel with the sharpest contrast gradient, at low frequencies in air and in tissues. Variability stems from differences in focal plane, insertion depth and sample thickness. | Standard error in manual tracking by an experienced user, thin tissue (e.g. pPSM, <100 μm) ±1.1 μm | Automatic tip position tracking using an algorithm reduces human errors. |
| Sample thickness or opaqueness | Imaging though thick tissue samples result in an increase of light scattering, deteriorates tip contrast and increases signal-to-noise ratio. | Standard error in thick tissues (e.g. closed neural tube, somites, >150 μm), ±2.2 μm | Use fluorescently labelled probes or surgically removing overlaying tissues can improve tip contrast. |
| High frequency movement | The resolution of high frequency probe movement is limited by the camera frame rate (up to 100 frames per second in this study). | Variable, up to ±4.8 μm | Increase imaging frequency. High frequency measurements are less relevant in tissue development contexts. |
| Lighting | Shadows/blight reflections cast by the light source can interfere with the algorithm's probe tracking. | Variable, up to ±5 pixels (3.0 μm) | Carefully ensure optimal lighting, confirm probe position manually if needed. Use the same lightening angles and intensity between control and samples. |
| Use of Epoxy-DiI or other labels on the probe | Use of Epoxy-DiI probe improves the signal-to-noise ratio and improves spatial coverage in thick tissues that obscure probe tips, such as in zebrafish early-stage embryo experiments. | Standard error (thick blastoderm and yolk) ±1.0 μm | Bright fluorescence light source and dim-light background. Sharp focus of the DiI boundary improves signal-to-noise ratio. |
| **Piezo movement** | | | |
| Voltage output | Use of the voltage controller interface results in a systematic error when setting voltage. | ±0.012 V | The addition of the interferometer provides reliable tracking of piezo position, allowing comparison across samples and removing the need to rely on voltage output. |
| Voltage to piezo movement relationship | Piezo movement when moving in 3 V intervals carry an error. In addition, the voltage-increase step size and voltage-decrease step size are different in certain ranges, data available from manufacturer. | Low voltages (<10 V): ±0.22 μm. Higher voltages (>30 V): ±3 μm | |
| Piezo drift | For long-term experiments piezo position can drift under no force or voltage input changes. | Over an hour, at 20 V drift is 5.4 μm ±3.5 μm. At 40 V drift is 30 μm (±22 μm) | |
| **Force estimation** | | | |
| AFM cantilevers | Manufacturers report a typical force constant range. The spring constant of TL-CONT cantilevers was calculated by the manufacturer using optical measurements of length, width and thickness. | Force constant possible range. AIO-TL: 0.04-0.7 N m$^{-1}$ ARROW-TL2: 0.004-0.54 N m$^{-1}$ HQ:CS38: 0.003-0.13 N m$^{-1}$ | Direct measurements of Eigen frequency and spring constant can be requested from the manufacturers. |
| Insertion depth | Insertion depth is controlled by recording the focal plane upon probe contact with the sample, and the desired focal plane. User experience and variability can affect the positioning of these planes. | Variable, up to ±4 μm | Consistent sample mounting, standard criteria for identifying features of the tissue at different depths. |
| Timing synchronisation | We record a timing delay between the interferometer and the camera. This is a hardware systems error that can be determined and corrected via calibration. | 0.6 s (±0.4 s) | The use of air controls to accurately quantify the timing delay is important for real-time force estimations. |

(quality>60), continue with automatic adjustment and wait until the software indicates that the sensor channel is valid. Next, set up the trigger signal required for the camera to activate the measurements to achieve synchronisation following the software instructions. Open the data stream monitor and assign the trigger. Set up the camera to send the trigger signal upon activating frames (i.e. image acquisition). Test the trigger by starting live stream of the camera, the interferometer signal should immediately show on the monitor plot.

With probe and interferometer stand-by, the control sequence can be performed, followed by the samples. For the control run, image an empty or water sample with the moving probe under the same voltage programme to be used on the embryo tissues, such as a step/ramp push for loading

**Table 3. Components of hydrogel substrates**

| Reagent | 79 Pa | 369 Pa | 1100 Pa | 3130 Pa | 5220 Pa | Source |
|---|---|---|---|---|---|---|
| 40% Acrylamide [ml] | 0.75 | 0.75 | 0.75 | 1 | 1.875 | Bio-Rad, 1610140 |
| 2% Bis-acrylamide [ml] | 0.18 | 0.3 | 0.5 | 1.125 | 0.375 | Bio-Rad, 1610142 |
| PBS [ml] | 9.005 | 8.885 | 8.75 | 7.875 | 7.685 | – |
| **To 1 ml add:** | | | | | | |
| TEMED [μl] | 15 | 15 | 1 | 1 | 15 | Thermo Fisher Scientific, 17919 |
| 10% Ammonium persulfate [μl] | 50 | 50 | 10 | 10 | 50 | Thermo Fisher Scientific, 50-995-775 |

experiments, or an oscillatory movement for rheology experiments. Before lifting the probe holder to allow mounting of the embryo sample, mark the position of the probe tip on the screen, as it will be difficult to see the probe tip directly before it making contacting the sample, once the sample is between the probe and the microscope. This mark helps when positioning the sample to the extract desired location of probe insertion. After mounting the embryo sample, adjust the focus to determine the location of the top surface of the sample, record the $z$-position of the surface and the probe tip $z$-position after insertion. These numbers help determine the insertion depth and probe-tissue contact area. This is not required for thin tissues, where the tissue thickness and probe width determine the contact area. To insert the probe into the sample, now without a clear visual of the probe tip, lower the probe holder in $z$ slowly and carefully while watching the changes on the live camera stream. The probe would change the lighting seen on the live stream and cast a shadow when it enters the tissue. Adjust the focus slightly to ensure a good contrast of the probe tip at the correct $z$-level of the tissue of interest. Depending on the experimental goals, a small region of interest for the camera and memory-based high frame-rate acquisition mode may be desired for capturing fast probe dynamics. In those cases, adjust exposure time and lighting angles/intensity to maximise contrast at high frame rates. Note that lower resolution, wider field of view images should also be taken to assess the range of tissue deformation. Collect the data obtained in a measurement run and record the corresponding measurement programme for downstream analysis.

### Consideration of measurement errors

The TiFM2.0 system offers significant improvements from TiFM1.0 in time synchronisation and probe positioning accuracy. The main error consideration remains the spatial accuracy of $x_T$, at the pixel with the sharpest contrast gradient with a resolution limit of $\sim 0.5$ μm in ideal conditions, producing the largest error term in low signal-to-noise conditions such as thick tissues, combining errors along the segmentation and tracking pipeline. Due to general hardware limitations, voltage output is not a reliable measure of probe movement; the use of the interferometer ($x_C$) marks a significant improvement in the tracking of piezo movement. Table 2 presents a list of noise sources and practices for minimising these sources of technical variability.

### Control samples

A semi-dilute polymer solution of 8.5% w/w, 13.5% w/w, 18.5% w/w and 23.5% w/w PIB (Sigma-Aldrich: 181455, H6703) was used. Hydrogel substrates were prepared following a protocol previously described (Tse and Engler, 2010; Xi et al., 2022), where the stiffness of the hydrogel solutions was directly measured using AFM. Briefly, 150 μl of the solutions described in Table 3 were sandwiched between two glass coverslips and left to polymerise for 20 min. The gels were hydrated in PBS for 30 min, after which one coverslip was carefully removed, exposing the hydrogel. The hydrogel was then immobilised on a plastic 35 mm dish using vacuum grease before TiFM measurements.

### Rheometer measurements of control samples

The rheological measurements of PIB solutions were conducted on a rheometer (Anton Paar MCR302) using a cone plate (50 mm in diameter, 1° angle) at 25°C. The shear strain was set to 6% with a gap of 20 μm, and the angular frequency was swept from 0.01 rad/s to 200 rad/s with 50 points per decade.

### Chicken embryos

Wild-type chicken (*Gallus gallus*) eggs were supplied by Medeggs Inc. Tg(CAG-GFP) (McGrew et al., 2008) and Tg(CAG-memGFP) (Rozbicki et al., 2015) chicken eggs were provided by the National Avian Research Facility (NARF) at the University of Edinburgh. Eggs were stored in a 15°C fridge and incubated in 37.5°C ~60% humidity incubators (Brisnea) for 24 h for the HH4 early-stage experiment, ~40 h to reach HH stage 10-12 for the body axis experiments, and 72 h for the blood pressure experiment. Embryos were prepared using a modified early chick (EC) culture (Chapman et al., 2001) protocol as described and incubated in a slide box lined with wet paper towels. Before TiFM experiments, embryos were further stabilised via a filter paper sandwich using enlarged windows to mitigate excess tension (Kunz et al., 2023) and mounted on a glass-bottom image dish with a thin layer of albumen or agarose-albumen culture media (Chan et al., 2023). A small amount of PBS or Ringer's solution was added on top of the embryo before probe insertion for short-term experiments. For longer-term experiments, the water compensation pipette and pump were used at a flow rate of 8 μl/min, with a solution of 70% dH₂O, 20% Ringer's, 10% thin albumen. When the heating insert is used, the outer ring temperature was set at 42°C to allow the dish centre to reach ~36°C at the embryo location.

### Zebrafish embryos

Wild-type zebrafish (*Danio rerio*) AB and TL strains were used. Embryos were kept in 1× Embryo medium (E3 medium, reagents included: NaCl, KCl, CaCl₂•H₂O, MgSO₄•7H₂O, HEPES, ddH₂O, pH to 7.4 with 10 M NaOH) at 28.5°C. After dechorionation, the embryos were kept in 1× E3 medium in a culture dish covered with agarose gel at 28.5°C. To hold the fish embryos and prevent rotations during measurements, a thin layer of 2% agarose was laid on the imaging dish, which was dug into with pipette tips to create two intersecting circles. E3 fish medium was then added to the imaging dish followed by loading the dechorionated embryo into one of the circles. The tissue to be tested is then rotated to face the opening between the two circles. The extra medium was then removed, leaving only a sufficient amount to cover the embryo. The probe was then inserted into the other circle, close to the test tissue. To ensure reliable tracking of the probe once it enters thicker tissue areas, a DiI-Epoxy bead was glued to the end of the cantilever to make a fluorescent trailing probe and fluorescence was imaged. All procedures involving embryos were regulated by the Animals (Scientific Procedures) Act 1986, Amendment Regulations 2012, and in compliance with ethical standards at the University of Cambridge.

### Data analysis

Movies were analysed in Fiji (ImageJ; Abramoff et al., 2004). To track the probe(s), the raw movies were cropped into small sections containing the range of the probe movement. Linear adjustments (brightness and contrast) were applied to sharpen the probe versus sample contrast. These processed movies were then used as input together with the meta information file (time stamps) and the interferometer data output file in a custom analysis script, to obtain the tracks of the probe(s) and the holder positions for downstream analysis. To estimate forces on the probe, deflection was determined by comparing the tracks in the sample with tracks in the control (no-resistance) run, as described in the text. In some cases, manual tracking of probe location was performed from the kymograph generated in Fiji. To approximately estimate tissue rheological properties, the standard material control data was extrapolated as a calibration curve to compare with amplitude measurements in the embryo. Expected probe amplitudes were adjusted by the spring constant, insertion depth and voltage depending on the experiment. For cell shape analysis, cell membranes were segmented using Cellpose (Stringer et al., 2021) and subsequently corrected manually.

### Acknowledgements

The authors thank Chon U Chan and members of the K. Kawaguchi lab for suggestions on the system design and data analysis; members of the Xiong lab for reagents, technical assistance and comments; Chun Yuan Hii and Clare Buckley for assistance with zebrafish embryos, which were obtained through the fish facility at the Department of Physiology, Development and Neuroscience of University of Cambridge; Charles Bradshaw (Gurdon institute) and James Steele (the Maxwell Centre of University of Cambridge) for 3D printing; Michela Geri and Gareth McKinley for sharing the OWCh software and suggestions; Chiu Fan Lee for suggestion of the mechanical propagation experiment.

### Competing interests

The authors declare no competing or financial interests.

### Author contributions

Conceptualization: A.R.H.-R., Y.L., F.J., F.X.; Data curation: A.R.H.-R., Y.L., S.B.P.M., J.M.N.V., R.L., Y.D., F.X.; Formal analysis: A.R.H.-R., Y.L., F.X.; Funding

acquisition: F.X.; Investigation: A.R.H.-R., Y.L., S.B.P.M., J.M.N.V., Y.D., F.X.; Methodology: A.R.H.-R., Y.L., F.J., S.B.P.M., J.M.N.V., R.L., L.D.M., L.B., F.X.; Project administration: F.X.; Resources: E.H., L.B., F.X.; Software: A.R.H.-R., Y.L., R.L., F.X.; Supervision: F.X.; Validation: A.R.H.-R., Y.L., F.X.; Writing – original draft: F.X.; Writing – review & editing: A.R.H.-R., Y.L., E.H., F.X.

**Funding**

This study is supported by a Wellcome Trust/Royal Society Sir Henry Dale Fellowship (215439/Z/19/Z) and a UK Research and Innovation-Engineering and Physical Sciences Research Council Frontier Research Grant (EP/X023761/1, originally selected as an ERC Starting Grant) to F.X. A.R.H.-R. acknowledges a School of Biological Sciences-Gurdon Studentship (University of Cambridge). Y.L. acknowledges a Cambridge Trust International Scholarship (University of Cambridge). F.J. and L.B. acknowledge a Herschel Smith Postdoctoral Fellowship (University of Cambridge). F.J. acknowledges a Wellcome Trust Early Career Award (302541/Z/23/Z). Y.D. acknowledges a Life Sciences Institute (Zhejiang University)-Gurdon Institute (University of Cambridge) exchange programme supported by China Scholarship Council. R.L. acknowledges a Xuetang summer scholarship from Tsinghua University and a Harding Distinguished Postgraduate Scholarship (University of Cambridge). L.D.M. acknowledges a Wellcome Trust PhD studentship (222274/Z/20/Z). Open Access funding provided by University of Cambridge. Deposited in PMC for immediate release.

**Data and resource availability**

All relevant data and details of resources can be found within the article and its supplementary information.

**Peer review history**

The peer review history is available online at https://journals.biologists.com/dev/lookup/doi/10.1242/dev.204549.reviewer-comments.pdf

**Special Issue**

This article is part of the Special Issue 'The Extracellular Environment in Development, Regeneration and Stem Cells', edited by Alex Hughes and Rashmi Priya. See related articles at https://journals.biologists.com/dev/issue/153/16

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
