## [Peer Review File · Development (Cambridge, England)]

TiFM2.0 - Versatile mechanical measurement and actuation in live embryos

Ana R. Hernandez-Rodriguez, Yisha Lan, Fengtong Ji, Susannah B.P. McLaren, Joana M. N. Vidigueira, Ruoheng Li, Yixin Dai, Emily Holmes, Lauren D. Moon, Lakshmi Balasubramaniam and Fengzhu Xiong
DOI: 10.1242/dev.204549

Editor: Thomas Lecuit

Review timeline

Original submission:	20 November 2024
Editorial decision:	14 February 2025
First revision received:	12 September 2025
Editorial decision:	28 October 2025
Second revision received:	23 November 2025
Accepted:	8 January 2026

Original submission

First decision letter

MS ID#: dev.204549

MS TITLE: TiFM2.0 - Versatile mechanical measurement and actuation in live embryos

AUTHORS: Ana R. Hernandez-Rodriguez, Yisha Lan, Fengtong Ji, Susannah B.P. McLaren, Joana M. N. Vidigueira, Ruoheng Li, Yixin Dai, Emily Holmes, Lauren D. Moon, Lakshmi Balasubramaniam and Fengzhu Xiong

Dear Dr Xiong,

I have now received all the referees' reports on the above manuscript, and have reached a decision. The referees' comments are appended below, or you can access them online: please go to:

As you will see, the referees express considerable interest in your work, but have some significant criticisms and recommend a substantial revision of your manuscript before we can consider publication. If you are able to revise the manuscript along the lines suggested, which may involve further experiments, I will be happy receive a revised version of the manuscript. Your revised paper will be re-reviewed by one or more of the original referees, and acceptance of your manuscript will depend on your addressing satisfactorily the reviewers' major concerns. Please also note that Development will normally permit only one round of major revision. If it would be helpful, you are welcome to contact us to discuss your revision in greater detail. Please send us a point-by-point response indicating your plans for addressing the referees' comments, and we will look over this and provide further guidance.

Please attend to all of the reviewers' comments and ensure that you clearly highlight all changes made in the revised manuscript. Please avoid using 'Tracked changes' in Word files as these are lost in PDF conversion. I should be grateful if you would also provide a point-by-point response detailing how you have dealt with the points raised by the reviewers in the 'Response to Reviewers' box. If you do not agree with any of their criticisms or suggestions please explain clearly why this is so.

Reviewer 1*Advance summary and potential significance to field*

This study focuses on the development of methods to probe the mechanical properties of developing tissues. The authors build on their previous work to upgrade tissue force microscopy. The authors highlight that this new approach reduced the holder footprint, while enhancing accessibility and the imaging signal. The paper is generally well-written and, in principle, developing methods to probe, and impose, mechanical forces in situ is of broad relevance, as this is a major bottleneck in the fields of mechanobiology and cell/developmental biology. I believe the study is of interest to a broad audience and represents a potentially important advance for the mechanobiology toolkit, however some revisions are needed, in my view, to support/clarify the findings in the manuscript and establish the reliability/full potential of the method. Given that my expertise is mainly in cell and developmental biology, I will concentrate my comments on the analyses and experiments that are needed in my view, to strengthen the key biological findings.

Comments for the author

1. The authors mention that "Combining these data, sample stresses and responses could be determined, and used to assess its mechanical properties quantitatively with finite element models if needed". This is an important step to establish TiFM as a method to not only impose mechanical deformations, but also directly measure forces and material properties. I appreciate that doing a full finite element method is out of scope, but believe that simple viscoelastic models (e.g. Kelvin) could readily be fitted to creep or relaxation curves, to at least get orders of magnitudes for viscoelastic time or elastic constant. In general, this should be done systematically for all mechanical assays and tissues provided in the manuscript and across embryos.
2. A calibration should be performed for the dual probe setup to validate the approach and guide the subsequent experiments/possible interpretations. It would be nice to indicate the detectable force range for the two setups and compared for low vs high frequencies. If force measurements are not possible with the dual setup in its current form, this should be made clearer and nonetheless strain rates, viscoelastic timescales and the dynamic deformation recovery from force application after probe retraction should be systematically analyzed (across samples and with statistics).
3. While I appreciate that the data shown in the Figures is representative, it is important to systematically include additional embryos in the supplementary material and quantitatively analyze all metrics across embryos (including with statistics). This is especially important, given that a key aspect of this work is establishing TiFM2.0 as a (improved) method to impose mechanical deformation and probe tissue mechanics in vivo.
4. What is the size of the probed regions across experiments (it is not always indicated)? And how does it relate with cell size in the different developmental timepoints and tissues?
5. It is challenging to observe the tearing and overall deformations highlighted by the authors in the Figures. Adding additional zoom-ins and/or more timepoints, with more detailed annotations in the figure and/or providing more supplementary videos would be helpful for the readers to follow the manuscript. Also, the time (e.g. before or after clamping) should be indicated in the figure directly.
6. Please include in the schematics in the figures both where the probes are introduced, but also which direction they are moved. Also, the indication of the anterior and posterior axis along with all relevant tissues discussed in the main text should be outlined in all schematics.
7. The authors show that the ultimate strain plateau reached is very similar for aPSM vs. pPSM (at least in this example), which seems inconsistent with the claim that the material properties in the two regions are different. Even the stiffness differences, as deduced from the strain at the amount

of stretch, are small (as acknowledged by the authors) - are these statistically significant or consistent across embryos at least?

8. While the authors highlight in the main text that deformation of tissues under the imposed forces can be analyzed in terms of cell shape and arrangement changes, this is clearer when performing these experiments in transgenic Tg(CAG:memGFP) embryos (as shown in Figure 2D for an earlier stage of development). Thus, to show this convincingly the authors could either include stepwise stretching experiments in aPSM vs. pPSM while imaging cell outlines in this genetic background or analyze the strain rate in their experiments in the epiblast. If this is not technically feasible/or doable in the timescale of revision, they could instead tone down this notion in the main text and raise it as a potential future direction, including the data in Figure 2d as supplementary.

9. The data for AP compression assays in the pPSM is not shown in the figures but still commented in the main text with a statement on differential relaxation following probe retraction: "Upon probe retraction after 5 min holding, the posterior tissues retain a significant amount of deformation.". This statement should be removed or the data included.

10. Please include the control experiment for Figure 4C and quantitatively compare the deformation following probe retraction in both cases.

11. For the statement "while increased plastic deformation is observed following persistent stress, the different deformability of the body axis between the anterior-posterior and medial lateral directions is consistent between short and long timescales." - the authors would need to repeat the experiments along the AP direction for longer timescales for this statement to be fully supported by data. Alternatively, they can remove the mention of anterior-posterior in this sentence.

12. The authors mention that "the resistance to probe movement measured could indicate the neighboring tissue connections and boundaries rather than local properties around the probe, which can be much stronger.", which is indeed intuitive. They mention the dual probe system can be used to circumvent this issue, but in their first set of experiments in the PSM they also observe micro-tearing in the aPSM (Figure 2). Can the authors comment further on how far these experiments can nonetheless still be used to calculate strain and deduce material properties?

13. A supplementary movie and snapshots should be provided for the tissue response in the experiment Figure 6h. Also, include a similar graph with the fit as in Figure 6e.

14. While potentially very interesting, the figure on zebrafish embryos seems still somewhat preliminary (e.g. the comparison between designs is not discussed; is a similar calibration needed, as epoxy probe was added? the n is not indicated in the figure legends). In my view, while a nice addition, it could be moved to supplementary and the claims adjusted in the main text.

Minor points:

1. Since aPSM and pPSM are indicated respectively in blue and red in the graphs, it would be helpful if the titles would reflect that color-code in Figure 2.

2. The authors should detail schematically in the figure how the strain was measured in Figures 2b/c. In figure 2c, the authors should also indicate in the figure directly that this analysis is performed following probe retraction. Additionally, it should be explicitly indicated that this data corresponds to retraction upon imposing the highest stretch (right?).

3. The stage of the animals used for the first part of Figure 2 (a-c) needs to be explicitly mentioned in the figure and figure legend, especially since it changes from the embryo stage shown in Figure 2d. More generally, this is often an important information missing from the figure legends (while its provided in the methods, it is easier for the reader to find this information directly in the figure legend).

4. All movies need to be called for in the main text also.

5. In all the supplementary movies, please indicate, or outline, the probe(s).

7. Can the authors add snapshots from this movie 3 in the figure? Also, in which direction is terms of embryo axes is the probe moved?

8. The color code in Figures 7d,e is very hard to follow and it would be nice to adjust it.

Reviewer 2

Advance summary and potential significance to field

In this article, Hernandez-Rodriguez et al., present an upgraded version of their Tissue Force microscopy (TFM1.0), which allowed to probe and perturb the mechanics of developing chicken embryos. The TIFM2.0 with its double probe, and interferometer-based detection of the imposed displacement represents a real improvement of the TIFM1.0 that now enables versatile measurements and perturbations, as showcased in the article. I find the setup and experiments very interesting, especially the one perturbing the embryo's mechanics. I think it would be a great tool for the community. However, I believe some aspects should be clarified prior to publication, especially regarding the way mechanical measurements are conducted which is superficially described.

Comments.

1. The description of how the system works is confusing and should be clarified in the text and the figures. This is important to understand how the measurements are made and interpreted (see point 2 below). The way the setup is described, it seems that upon actuation, the tip and holder undergo a uniaxial movement (x). It took me quite some time to figure out (mainly by looking up the reference of the piezo in the methods) that the piezo used in this study is not a uniaxial piezo motor but a piezo bender provoking a movement in both x and z axes. The authors should clarify this in the text (e.g. replace the wording piezo motor by piezo bender) and in Figure 1 by showing the rotation of the probe and holder on the xz axis upon actuation.

2. Related to this point, why the authors show raw data for displacement of tip and holder in Figure 1E is really unclear. From the text it is difficult to understand why in their control (without resistance) a difference is observed between the tip and holder position in Figure 1E. The authors should show their data after their calibration has been used to correct for the displacement in z of the cantilever, thereby revealing only its deflection, which is the meaningful information. The way the data is presented in Figure 1E, it seems that the system does not work, measuring a deflection without resistance. The raw data and the linear relationship between tip and holder position (claimed by the authors) should be presented supplementary figures for all the experiments, such that the reader has an idea of how the different the calibration is from one experiment to another. The number of time this experiment has been performed should also be included.

3. Out of curiosity, why don't the authors use a uniaxial piezo motor whose displacement is known and controlled? wouldn't that eliminate the need for the interferometer?

4. Regarding the experiment in which the anterior and posterior PSM are stretched:

- Can the authors provide the movie ? that would allow the reader to visually compare the different responses?
- Were posterior and anterior measurements performed in the same embryo?
- Because the authors measure strains and not mechanical properties I suggest that they be more cautious when claiming that "These resultst show that the aPSM is stiffer and more viscous than the pPSM" .
- Regarding Figure 2B. What is the variability between embryos? Can the authors show the data for all 5 experiments in the supplementary? Why the imposed displacement steps don' t have the same duration between aPMS and pPSM? While neither of them really reach asymptote? With equal duration it would be easier to compare their response.
- Regarding Figure 2C. The authors mention that "Upon probe retrieval after 5 minutes of holding, the tissues (now tracked manually by the insertion wound sites) both retracted, with the aPSM shortening faster and to a larger extent over time". This is not obvious from the figure, representing 1 experiment (?). How variable is the result from embryo to embryo? Is the difference

significant? Can the authors fit a simple model, obtain the initial relaxation speed and do statistical tests using this? Again, an average or a plot showing all Ns would allow the reader to appreciate how reproducible the measurements are, especially since mechanical measurements are notoriously variable from embryo to embryo.

5. Regarding Figure 3A :

- What do the authors mean when they mention " that the probes are calibrated through synchronized control"? is it in air, without resistance?
- P(R) and P(L) are not defined.

6. Regarding Figure 4:

- The term "retraction" of the probe is not defined, I understand from the figure that it refers to decreasing the voltage, returning the holder to its initial position. But this should be defined as it could be confused with the "retrieval" of the probe, as in Fig2.
- The authors introduce the notion of plastic deformation, can they define it in particular in relation to viscous deformation?

7. Regarding data in figure 5, axes in Figure 5F are not labeled.

8. Regarding Data in Figure 6:

- Is the probe vertical, as shown in Figure 6A? if so this should be mentioned in the text. Are the two probes in Figure 6H also vertical? Is the calibration with no resistance performed prior to measurement? or there is no calibration anymore?
- I have a similar comment that the one in point 2. Why don't the authors display the calibrated deflection of the cantilever rather than the position of chip and Tip? Again, aren't calibrated deflections the most informative information?
- The phase shift is not obvious in Figure 6B.
- There is no conclusion drawn from the result from figure 6C, regarding the relation between magnitude depth and frequency.
- In Figure 6E, can the authors show here again the calibrated deflection instead of only the displacement of the tip?

9. Regarding Figure 7

- Same comments for previous figures, regarding the plotting of the calibrated deflection. It is unclear if the interferometer is still even used in these experiments.

Reviewer 3

Advance summary and potential significance to field

The manuscript presents TiFM2.0, an upgraded tissue force microscopy system designed for live embryonic mechanical measurements. By incorporating interferometer positioning and a dual-probe configuration, the system allows for improved imaging accessibility and a wider range of mechanical perturbations, including bidirectional stretching, compression, and stress propagation experiments. The authors provide a set of proof-of-concept applications in chicken and zebrafish embryos, demonstrating the system's capability to measure and manipulate tissue deformation with precision. They also include simplified protocols to facilitate the system's adoption in other developmental biology laboratories.

The study looks technically robust, and the advancements over TiFM1.0 are clearly articulated. The ability to perform real-time, minimally invasive force measurements is a significant contribution to the field of morphogenesis and developmental mechanics. However, while the method is presented as a tool for stress and force measurements, there is no presentation of actual force-stress data, nor are the tissue deformations measured interpreted in terms of mechanical parameters (e.g., force vs. displacement curves, dynamic moduli). Additionally, the interpretation of stress as a scalar rather than a tensor requires clarification. Finally, the manuscript lacks a quantitative assessment of inherent system errors and calibration uncertainties, and insufficient statistical analysis of the presented data.

Comments for the author

(1) Lack of error estimation and calibration uncertainty

While the authors introduce interferometer positioning for improved precision, the manuscript does not provide a quantitative estimation of errors inherent to the system or potential calibration uncertainties. Addressing these issues is critical, as small variations in probe alignment, positioning drift, or sample deformation artifacts may introduce measurement errors.

Suggested improvements:

- Provide an explicit quantification of the measurement errors associated with probe positioning, force estimation, and strain measurements.
- Conduct control tests of the system using well-characterized synthetic material with known viscoelastic properties.
- Discuss potential sources of noise (e.g., environmental factors, humidity sensitivity of capacitors in TiFM1.0) and how they have been mitigated in TiFM2.0.

(2) Insufficient presentation of force or stress measurements

Although the manuscript presents TiFM2.0 as a mechanical measurement tool, there is a lack of explicit force or stress data: unless I missed a plot, all figures present measured displacements but never their translation into mechanically explicit information (which generally requires an explicit force-displacement measure and a mechanical model), such as Young's modulus, viscoelastic relaxation timescales, storage or loss moduli. The authors describe mechanical perturbations and tissue responses qualitatively but do not provide direct force-displacement or stress-strain curves. Specific concerns:

- The stress formulation presented, $\sigma = k \cos(\theta) [f(x_{C1}) - x_{T1}] w^{-1} (z - z_0)^{-1}$, suggests a scalar representation rather than a full stress tensor. However, biological tissues exhibit anisotropic and spatially heterogeneous mechanical properties.
- There seems not to be clear distinction between force and stress measurements, which could lead to misinterpretation of the mechanical properties being assessed.

Suggested improvements:

- Provide explicit force-displacement curves for different tissue types to illustrate mechanical responses under stretching and compression.
- If possible, introduce stress-strain relationships or at least discuss why such characterization was not performed.
- Clarify whether the force measurements correspond to local point forces or averaged tissue stresses, and justify the chosen approach.
- Discuss whether tensorial stress components could be extracted in future studies.

(3) Limited quantitative analysis of mechanical properties

(3.1) Limited quantitative analysis of measurements

The paper would benefit from a more extensive mechanical characterization of tissues beyond qualitative descriptions of deformation. Standard rheological measurements, such as storage (G') and loss (G'') moduli in oscillatory forcing experiments, are absent (Fig. 6).

Suggested improvements:

- Provide at minimum viscoelastic timescale estimations by fitting relaxation curves with single or double exponential functions (Fig. 1E-F, Fig. 2B, Fig. 3A, Fig. 4A-C).
- In Fig. 6, perform oscillatory mechanical testing at varying frequencies to determine the full spectral viscoelastic response of embryonic tissues, extracting G' and G'' as a function of forcing frequency.
- Compare measured forces and mechanical moduli with some known values from previous studies (relates to Point 1, control tests).

(3.2) Insufficient statistical analysis

While the study presents measurements across different conditions, the statistical analysis is limited, and the manuscript lacks a detailed quantification of measurement variability across biological replicates. This is particularly critical for a biomechanical study, where sample-to-sample variation could be significant.

Suggested improvements:

- Report mean values with standard deviations or confidence intervals for key measured mechanical parameters (points 2 and 3.1).
- Provide sample sizes (N) for each measurement condition and include statistical tests.

Minor comments:

- Figure 1: The schematic of the TiFM2.0 setup could be enhanced by providing a clearer comparison to TiFM1.0, highlighting the key technical improvements.
- Figure 2: Tissue stretching experiments would be strengthened by additional kinematic tracking of cell shape deformations.
- The term "stress" should be used carefully to avoid confusion with "force," particularly when presenting data in biological contexts.

Conclusion

This study presents a valuable technological advancement in the field of live embryo mechanics. TiFM2.0 offers improved precision and versatility compared to its predecessor, and its ability to perform bidirectional perturbations is a significant step forward. However, the manuscript would benefit from a more rigorous error analysis, clearer distinction between stress and force measurements, and a more thorough analysis of tissue mechanical properties. Addressing these concerns could significantly enhance the impact and reproducibility of the findings.

First revision

Author response to reviewers' comments

Hernandez et al., response to reviewer comments

Reviewer 1: SUMMARY OF THE ADVANCE MADE IN THIS PAPER AND ITS POTENTIAL SIGNIFICANCE TO THE FIELD

This study focuses on the development of methods to probe the mechanical properties of developing tissues. The authors build on their previous work to upgrade tissue force microscopy. The authors highlight that this new approach reduced the holder footprint, while enhancing accessibility and the imaging signal. The paper is generally well-written and, in principle, developing methods to probe, and impose, mechanical forces in situ is of broad relevance, as this is a major bottleneck in the fields of mechanobiology and cell/developmental biology. I believe the study is of interest to a broad audience and represents a potentially important advance for the mechanobiology toolkit, however some revisions are needed, in my view, to support/clarify the findings in the manuscript and establish the reliability/full potential of the method. Given that my expertise is mainly in cell and developmental biology, I will concentrate my comments on the analyses and experiments that are needed in my view, to strengthen the key biological findings.

We thank this reviewer for their thorough, critical and constructive comments.

SUGGESTIONS TO AUTHORS

1. The authors mention that "Combining these data, sample stresses and responses could be determined, and used to assess its mechanical properties quantitatively with finite element models if needed". This is an important step to establish TiFM as a method to not only impose mechanical deformations, but also directly measure forces and material properties. I appreciate that doing a full finite element method is out of scope, but believe that simple viscoelastic models (e.g. Kelvin) could readily be fitted to creep or relaxation curves, to at least get orders of magnitudes for viscoelastic time or elastic constant. In general, this should be done systematically for all mechanical assays and tissues provided in the manuscript and across embryos.

In the revision we provide results of multiple embryos and simple viscoelastic model fitting as suggested by the reviewer. We also discuss the limitations of simple models and emphasize the challenges of the complex, multi-tissue environment - thus the caution that one must take while

interpreting TiFM results and results using other approaches. Some example applications are being developed/incorporated into their respective in-depth studies and not yet ready to be presented as conclusive new biological insights in this manuscript, which primarily aims at sharing the hardware and protocols as a resource for the community.

Specifically:

New Figure 2D-H. We fitted the deformation recovery phase of the PSM across multiple embryos after probe retrieval using a two-term exponential model which captured the fast (1-5 seconds) and the slow (20-50 seconds) timescales. The model contains a residual strain term that is not accounted for by the generalized SLS model as a result of tissue plasticity. The results show variability of max and residual strains across embryos, yet consistently higher for the posterior PSM than the anterior PSM. The slow relaxation timescale is smaller for the posterior PSM, suggesting a more compliant tissue. These simple fittings provide useful quantifications on the differences between tissue locations, but also show that simple viscoelastic models are insufficient to capture all aspects of the tissue behaviour observed.

New Figure 5C-G (previous Figure 4). We added multiple embryos and tracked their deformation and recovery. The recovery kymographs were fitted with an exponential term and the tissue displacements and the timescale of recovery were compared across embryos and between close and far tissue locations. The results show an enhanced deformability along the AP axis at the posterior progenitor domain near the compressive probe, but not along the ML axis.

New Figure 5H-I. For the long-term culture and bending experiment along the ML axis, we now included a baseline collection of probe stability control, and tested embryos under different extent of bending. A direct comparison example of short vs long term recovery dynamics is shown in 5I, showing the long-term plastic deformation. As the long-term stretching experiments along the AP axis have been well-documented in Chan et al., 2023, we cited those results here.

New Figure 7A-C. To test TiFM's capacity in mechanical property measurements, we performed new experiments with different concentrations of the standard viscoelastic material PIB. These materials were control-tested by a rheometer (**New Figure S1I**). The control values allowed us to compare it with the TiFM probe amplitudes generated both locally and across the sample at the second probe. We found that in a given setting (probe stiffness, depth and frequency) the probe amplitudes consistently and sensitively detect the differences across 3 orders of magnitude of the storage modulus of the material (7B(ii)) and are also sensitive to the frequency domain (7C(ii)). These results, together with the solid (hydrogel) results (Figure 7D), demonstrate that the probes have the resolution needed to measure the embryonic tissue properties which fall within this range. *While one may be tempted to use these data as calibration and then assign a modulus to the in vivo tissue measurements we performed, we refrain from doing that as the heterogeneity of small-tissue embryo environment is fundamentally different from simple polymer materials.*

We agree with the reviewer that quantitative interpretation of TiFM data in terms of material properties is a key future goal with carefully developed, validated models (such as a full finite element model). We are currently creating a COMSOL model incorporating the probe geometry and measurement dynamics to recapitulate the measurement of complex modulus in control materials, as the first step. We will then extend this model by supplying inhomogeneous samples, such as two connected but distinct tissues (e.g., PSM and neural tube) where assumptions are made about their material properties and their ECM connections. These will then be fit with TiFM experimental data in and between these tissues. These steps are significant work and non-trivial, not possible to achieve in the current revision time frame.

Related to this point, a recent preprint (<https://doi.org/10.1101/2025.05.11.653307>) by the groups of Francis Corson and Jerome Gros presented a finite element model to interpret the mechanical properties of the quail blastoderm under a cantilever-driven deformation with an adhesive bead. This work sets up a great example of data fitting that we are learning from. The reviewers can gauge the work involved in the strain field measurement and the model fitting process in this work. Significant changes to the modeling will be required for each of the versatile experimental settings presented in our work.

The above points are discussed in the revision.

2. A calibration should be performed for the dual probe setup to validate the approach and guide the subsequent experiments/possible interpretations. It would be nice to indicate the detectable force range for the two setups and compared for low vs high frequencies. If force measurements are not possible with the dual setup in its current form, this should be made clearer and nonetheless strain rates, viscoelastic timescales and the dynamic deformation recovery from force application after probe retraction should be systematically analyzed (across samples and with statistics).

As shown in Figure 4 (previous Figure 3), we perform a control movement sequence of the dual probes to obtain their distance dynamics under no-load, to then estimate the force in an experimental run when the probes encounter the tissue load. We rely on the commercially supplied AFM cantilevers (which can be individually calibrated to obtain an exact spring constant by the manufacturer) and do not have an experimental set-up (something similar to a micro-spring with the cross-section size of the probes) to directly validate the forces. We do however, have performed dual probe actuation of PIB polymers at different frequencies (**New Figures 7B-C**) which help validate that the deflection of the probes (amplitude changes) corresponds to the order of magnitude of forces imposed on the known materials. However, an accurate quantitative calibration will require a model that accounts for the probe geometry and the 3D strain field of the sample which are not yet available.

The detectable force range depends on the properties of the probes used. The lower limit is defined by the accuracy of probe tip tracking, which is sample-dependent (e.g., thicker, less transparent tissues make tip tracking less accurate), currently at $\sim 1\mu\text{m}$, the softest probe we use is 0.02N/m , setting the lower limit at 20nN . The upper limit would depend on the strength of the stiffer probes, which would far exceed the forces embryonic tissue produce. This point has been added to the main text as the following: *“Similar to TiFM1.0, the theoretical force sensitivity limit is at the order of 10nN , set by the resolution of probe tip tracking currently at the order of $1\mu\text{m}$ and the softest probes we tested at the order of 0.01N/m . In practice, we detect most embryonic tissue forces between 100nN and $10\mu\text{N}$ (tissues yield at the higher end).”*

Quantitative data across samples and with statistics have been added for variable imposed strain rates (by frequency modulation, **new Figure 7C**), viscoelastic timescales and dynamic deformation recovery from force application after probe retraction (**new Figures 2D-H, 5C-I**).

In addition, we provided a table of error sources in Materials & Methods section, magnitudes and mitigation to help readers estimate the sensitivity and ranges for specific samples and applications.

3. While I appreciate that the data shown in the Figures is representative, it is important to systematically include additional embryos in the supplementary material and quantitatively analyze all metrics across embryos (including with statistics). This is especially important, given that a key aspect of this work is establishing TiFM2.0 as a (improved) method to impose mechanical deformation and probe tissue mechanics in vivo.

We agree. Additional samples have been analysed to show the variability in the example measurements, and statistics added where appropriate. Specifically:

New Figure 1. Illustration is improved and a clearer representative example of measurement principle is included (with additional replicates shown in **New Figure S1**). No biological conclusion is made here.

New Figure 2. D-H, a group of 5 embryos were fitted and their fast and slow relaxation timescales were compared.

New Figure 3. This expands the analysis of cellular changes under the TiFM double probe stretching experiment. A large number of cells were segmented and compared between 2 regions of the movie. Cell shapes and orientations were compared systematically with statistics. This movie and analysis are representative of other repeats.

New Figure 4. Additional samples added and the resistance in compressive and extensive directions were compared.

New Figure 5. Additional samples added, fitted, and compared several metrics statistically.

New Figure 6. Presentation is improved with labels. Variability of blood pressure estimation is noted in the main text.

New Figure 7. New range of PIB samples added with statistical tests. All results in this Figure now have multiple samples with statistics.

New Figure 8. Additional samples added and the results statistically analyzed.

4. What is the size of the probed regions across experiments (it is not always indicated)? And how does it relate with cell size in the different developmental timepoints and tissues?

The images are labelled with scalebars and the distances (e.g., between 2 probes) are indicated where appropriate so the readers are well informed. The embryo illustrations are also proportional to the scale of the images shown. The cell sizes are $\sim 10\mu\text{m}$ in these stages and are generally one to two orders of magnitude smaller than the size of the probed regions (variable from 100-1000 μm).

This exact size of tissue that is impacted by the probe has implications on data interpretation (e.g., determining the strain). It is not immediately clear how far the mechanical impact of the probe reaches in the complex *in vivo*, *in situ* context. The size of probed regions in simple materials such as the hydrogel and PIB can be defined by tracker particles or the second probe (new Figure 7B(i),C(i)). In the embryonic tissue, comprehensive imaging will be required to map the strain fields around the probe. In a solid tissue like the notochord, the actuation could propagate far along the notochord. In a fluid tissue like the posterior PSM, the actuation may drop quickly near the probe (e.g., Figure 7H, Movie S7). Still, as the tissue mechanical properties are complex with multiple structural components, an elastic component could propagate a small part of the probe actuation in the posterior PSM to far distances. We are actively investigating these questions by performing systematic mapping with the TiFM. The findings will be reported in future manuscripts.

5. It is challenging to observe the tearing and overall deformations highlighted by the authors in the Figures. Adding additional zoom-ins and/or more timepoints, with more detailed annotations in the figure and/or providing more supplementary videos would be helpful for the readers to follow the manuscript. Also, the time (e.g. before or after clamping) should be indicated in the figure directly.

We added an annotated zoom-in for the tearing (**New Figure 2A''' and 2A''''**). We also included **New Supplemental Movies** for the samples shown in Figure 2A. We also added a time axis showing the clamping timeline in the **New Figures 6E-F**.

6. Please include in the schematics in the figures both where the probes are introduced, but also which direction they are moved. Also, the indication of the anterior and posterior axis along with all relevant tissues discussed in the main text should be outlined in all schematics.

We added arrows in all schematics to show the direction of probe movement, and the anterior-posterior axis where relevant. Relevant tissues were labelled in the schematics and on figures, and abbreviations spelt out in the figure legends.

In the schematics where a full embryo illustration was shown with the head and the body, we find it unnecessary to label the A-P axis.

7. The authors show that the ultimate strain plateau reached is very similar for aPSM vs. pPSM (at least in this example), which seems inconsistent with the claim that the material properties in the two regions are different. Even the stiffness differences, as deduced from the strain at the amount of stretch, are small (as acknowledged by the authors) - are these statistically significant or consistent across embryos at least?

Indeed there is variability here. In the **New Figure 2D-H**. We fitted the deformation recovery phase of the PSM across multiple embryos after probe retrieval using a two-term exponential model which captured the fast (1-5 seconds) and the slow (20-50 seconds) timescales. The model contains a residual strain term that is not accounted for by the generalized SLS model as a result of tissue plasticity. The results show variability of max and residual strains across embryos, yet consistently higher for the posterior PSM than the anterior PSM. The slow relaxation timescale is smaller for the posterior PSM, suggesting a more compliant tissue. These simple fittings provide useful quantifications on the differences between tissue locations, but also show that simple viscoelastic models are insufficient to capture all aspects of the tissue behaviour observed.

Regarding the plateau/max strain reached, the 5 samples show variable differences between aPSM and pPSM (**New Figure 2E**). Part of these variabilities could be of technical origin (such as probe insertion depth difference between different embryos), not necessarily all biological. We describe practices for minimizing the technical variabilities of the experiments in the methods section of the revision.

8. While the authors highlight in the main text that deformation of tissues under the imposed forces can be analyzed in terms of cell shape and arrangement changes, this is clearer when performing these experiments in transgenic Tg(CAG:memGFP) embryos (as shown in Figure 2D for an earlier stage of development). Thus, to show this convincingly the authors could either include stepwise stretching experiments in aPSM vs. pPSM while imaging cell outlines in this genetic background or analyze the strain rate in their experiments in the epiblast. If this is not technically feasible/or doable in the timescale of revision, they could instead tone down this notion in the main text and raise it as a potential future direction, including the data in Figure 2d as supplementary.

In the **New Figure 3**, expanded from original Figure 2D, we provide an analysis of populational cellular shape changes and tracking of individual cells over different time points, using segmentation of the apical surface of epiblast cells by Cellpose and manual correction. We found that cells located in the central region between the probes show more pronounced size increase and shape changes than those of cells in the nearby off-centre region (Figures 3A-D), consistent with the region between two probes experiencing maximum tension. The shape changes show an average elongation along the direction of imposed tension (Figure 3C). To examine the dynamics of single cells during stretching, we tracked ten individual cells over time, which revealed that cell shape parameters progressively changed with variations upon stretching. These include an average apical size increase (Figure 3E), re-orientation of the long axis (Figure 3F), and alignment towards the direction of imposed tension (Figure 3G). These results show that TiFM2.0 enables cellular dynamics analysis under mechanical perturbations of the tissue.

9. The data for AP compression assays in the pPSM is not shown in the figures but still commented in the main text with a statement on differential relaxation following probe retraction: "Upon probe retraction after 5 min holding, the posterior tissues retain a significant amount of deformation.". This statement should be removed or the data included.

In the **New Figure 5C-G**, these data are quantified with multiple samples.

10. Please include the control experiment for Figure 4C and quantitatively compare the deformation following probe retraction in both cases.

In the **New Figure 5H-I**, a repeat experiment systematically including different controls is presented and the text description adjusted accordingly.

11. For the statement "while increased plastic deformation is observed following persistent stress, the different deformability of the body axis between the anterior-posterior and medial lateral directions is consistent between short and long timescales." - the authors would need to repeat the experiments along the AP direction for longer timescales for this statement to be fully supported by data. Alternatively, they can remove the mention of anterior-posterior in this sentence.

This experiment was well documented in our previous paper (Chan et al., 2023). We have clarified this in the results section. The statement has been revised to prevent confusion.

12. The authors mention that "the resistance to probe movement measured could indicate the neighboring tissue connections and boundaries rather than local properties around the probe, which can be much stronger.", which is indeed intuitive. They mention the dual probe system can be used to circumvent this issue, but in their first set of experiments in the PSM they also observe micro-tearing in the aPSM (Figure 2). Can the authors comment further on how far these experiments can nonetheless still be used to calculate strain and deduce material properties?

The dual probe system is helpful in the sense that they can cancel out the net force on the whole embryo, preventing for example a global drift from a distant location. The dual probes also provide the mechanical propagation experiments testing primarily the tissue between the probes (Figure 7H). So in principle, large stresses like those causing tearing in the aPSM can be avoided in measuring material properties.

In the **New Figures 7A-C**. To test TiFM's capacity in mechanical property measurements, we performed new experiments with different concentrations of the standard viscoelastic material PIB. These materials were control-tested by a rheometer (**New Figure S11**). The control values allowed us to compare it with the TiFM probe amplitudes generated both locally and across the sample at the second probe. We found that in a given setting (probe stiffness, depth and frequency) the probe amplitudes consistently and sensitively detect the differences across 3 orders of magnitude of the storage modulus of the material (7B(ii)) and are also sensitive to the frequency domain (7C(ii)). These results, together with the solid (hydrogel) results (Figure 7D), demonstrate that the probes have the resolution needed to measure the embryonic tissue properties which fall within this range. *While one may be tempted to use these data as calibration and then assign a modulus to the in vivo tissue measurements we performed, we refrain from doing that as the heterogeneity of small-tissue embryo environment is fundamentally different from simple polymer materials.*

We agree with the reviewer that quantitative interpretation of TiFM data in terms of material properties is a key future goal with carefully developed, validated models (such as a full finite element model). We are currently creating a COMSOL model incorporating the probe geometry and measurement dynamics to recapitulate the measurement of complex modulus in control materials, as the first step. We will then extend this model by supplying inhomogeneous samples, such as two connected but distinct tissues (e.g., PSM and neural tube) where assumptions are made about their material properties and their ECM connections. These will then be fit with TiFM experimental data in and between these tissues. These steps are significant work and non-trivial, not possible to achieve in the current revision time frame.

Related to this point, a recent preprint (<https://doi.org/10.1101/2025.05.11.653307>) by the groups of Francis Corson and Jerome Gros presented a finite element model to interpret the mechanical properties of the quail blastoderm under a cantilever-driven deformation with an adhesive bead. This work sets up a great example of data fitting that we are learning from. The reviewers can gauge the work involved in the strain field measurement and the model fitting process in this work. Significant changes to the modeling will be required for each of the versatile experimental settings presented in our work.

The above points are discussed in the revision.

13. A supplementary movie and snapshots should be provided for the tissue response in the experiment Figure 6h. Also, include a similar graph with the fit as in Figure 6e.

A New Supplementary movie (Movie S7) is provided as suggested. For this experiment, the amplitude was measured on the kymograph directly and not by a fit as in the other figure (in embryo measurements, the tip tracking is noisier, requiring fitting).

14. While potentially very interesting, the figure on zebrafish embryos seems still somewhat preliminary (e.g. the comparison between designs is not discussed; is a similar calibration needed, as epoxy probe was added? the n is not indicated in the figure legends). In my view, while a nice addition, it could be moved to supplementary and the claims adjusted in the main text.

The main goal of this manuscript is to showcase diverse examples of possible applications of the system, maximizing community interest.

In the **New Figure 8D-F**, we repeated the experiment with a larger number of embryos. These results clearly establish that the dorsal trunk of the older fish embryo is stiffer than the blastoderm of early embryos. It's possible that the epoxy may change the stiffness of the probes, while the relative differences detected using the same probe remain interesting information.

Minor points:

1. Since aPSM and pPSM are indicated respectively in blue and red in the graphs, it would be helpful if the titles would reflect that color-code in Figure 2.

Updated this on Figure 2A.

2. The authors should detail schematically in the figure how the strain was measured in Figures 2b/c. In figure 2c, the authors should also indicate in the figure directly that this analysis is performed following probe retraction. Additionally, it should be explicitly indicated that this data corresponds to retraction upon imposing the highest stretch (right?).

The strain is calculated as the distance between the two probes over the distance at time=0, minus 1. We added figure titles to the new Figure 2B-C to make this clear. We clarified in the text that the retrieval follows the highest stretch at a consistent time (5min) for different embryos.

3. The stage of the animals used for the first part of Figure 2 (a-c) needs to be explicitly mentioned in the figure and figure legend, especially since it changes from the embryo stage shown in Figure 2d. More generally, this is often an important information missing from the figure legends (while its provided in the methods, it is easier for the reader to find this information directly in the figure legend).

We agree! HH stage information has been added throughout Figure legends for all the data.

4. All movies need to be called for in the main text also.

Yes.

5. In all the supplementary movies, please indicate, or outline, the probe(s).

We prefer not to block dynamic changes around the probe by additional annotations over the movies. The probes are high-contrast and distinctly visible on the movies. The probe locations are explained to the reader in the movie legends.

7. Can the authors add snapshots from this movie 3 in the figure? Also, in which direction in terms of embryo axes is the probe moved?

We apologize for the limited space on Figure 7. This movie corresponds to Figure 7E where the probe is moving along the A-P axis. This has been noted in the movie legend.

8. The color code in Figures 7d,e is very hard to follow and it would be nice to adjust it.

We apologize for the quality of presentation in the initial version. This has been replaced by new repeats with a large number of embryos, including distinct color codes.

Reviewer 2: In this article, Hernandez-Rodriguez et al., present an upgraded version of their Tissue Force microscopy (TFM1.0), which allowed to probe and perturb the mechanics of developing chicken embryos. The TIFM2.0 with its double probe, and interferometer-based detection of the imposed displacement represents a real improvement of the TIFM1.0 that now enables versatile measurements and perturbations, as showcased in the article. I find the setup and experiments very interesting, especially the one perturbing the embryo's mechanics. I think it would be a great tool for the community. However, I believe some aspects should be clarified prior to publication, especially regarding the way mechanical measurements are conducted which is superficially described.

We thank this reviewer for their thorough, critical and constructive comments.

Comments.

1. The description of how the system works is confusing and should be clarified in the text and the figures. This is important to understand how the measurements are made and interpreted (see point 2 below). The way the setup is described, it seems that upon actuation, the tip and holder undergo a uniaxial movement (x). It took me quite some time to figure out (mainly by looking up the reference of the piezo in the methods) that the piezo used in this study is not a uniaxial piezo motor but a piezo bender provoking a movement in both x and z axes. The authors should clarify this in the text (e.g. replace the wording piezo motor by piezo bender) and in Figure 1 by showing the rotation of the probe and holder on the xz axis upon actuation.

Thanks for this suggestion. We have clarified the piezo mechanism with a clear schematic in the **New Figure 1B(iii)** (exaggerated in terms of the amount of the rotation/bending), and **New Figure 1E-G** to clearly explain what happens in a measurement.

Because of the extremely small angle of bending (~0.2 degrees at 100µm of x displacement), in combination with the significant length of the piezo + the extension probe chip holder + the length of the chip and cantilever (~7mm), the z axis movement in the piezo bender is about >3 orders of magnitude smaller than the x axis movement (~0.04µm in z per 100µm in x). Therefore, the probe movement is uniaxial in x for practical purposes (errors brought in by z shifts are orders smaller than other errors, such as the segmentation errors of the probe tip). Manufacturer's and our illustrations (**New Figure 1B(iii)**) will exaggerate the bending nature of the mechanism because realistic illustrations will make it impossible to see the z axis movement.

2. Related to this point, why the authors show raw data for displacement of tip and holder in Figure 1E is really unclear. From the text it is difficult to understand why in their control (without resistance) a difference is observed between the tip and holder position in Figure 1E. The authors should show their data after their calibration has been used to correct for the displacement in z of the cantilever, thereby revealing only its deflection, which is the meaningful information. The way the data is presented in Figure 1E, it seems that the system does not work, measuring a deflection without resistance. The raw data and the linear relationship between tip and holder position (claimed by the authors) should be presented supplementary figures for all the experiments, such that the reader has an idea of how the different the calibration is from one experiment to another. The number of time this experiment has been performed should also be included.

We apologize for the confusion. The X_C and X_T measurements are not meant to be identical as the measurements are taken at different heights of the whole piezo-probe construct. Now clearly illustrated with the **New Figure 1B(iii)**, there is a variable distance between the two measurement locations (L_{CT}), which is changeable depending on probe and holder choices of the specific experiment. Because of the rotational nature of the piezo, X_T will increase more than X_C as the system moves in one direction, at a ratio given by $(L_{CT}+LP)/LP$, where LP is the length of the piezo. This effect is shown in **New Figure 1E**, no loading control, where we picked a particularly long holder (large L_{CT}) to show the difference. This control provides the relationship between X_C and X_T in a zero-deflection condition. Subsequently, in a sample, as shown in **New Figure 1F**, the probe displacement is greatly reduced by the resistance of the sample. The force is then calculated by comparing the change in X_T at a given X_C . The calculated forces are shown in the **New Figure 1G**. Additional examples can be found in **New Supplemental Figure 1A-F**.

The no-load calibration (air or water control) is performed for each holder/probe combination. A new no-load control is run whenever the probe or holder is changed. We have included the no-load sequence where forces are quantified. While the relationship between X_C and X_T is mostly linear, we do not use a linear model to perform force calculation but rather use the matching X_T from the no-load sequence to ensure accurate estimate of deflection.

In the revision, we have added additional samples and indicated n where applicable throughout the manuscript.

3. Out of curiosity, why don't the authors use a uniaxial piezo motor whose displacement is known and controlled? wouldn't that eliminate the need for the interferometer?

We are certainly open to better hardware options, if the reviewer has specific suggestions we'd be happy to evaluate them for future designs. To our knowledge, the precision of displacement and its control vary greatly among available linear stages and actuators. The most common screw-based and belt-based stages have good ranges but are limited in precision and speed variability (e.g., light microscopes require image stitching algorithms after a tile scan, as the stages are limited in their precision). The other side of this is the controller/encoder. Those using electromagnetic controls require more specific control systems and a large footprint. Stages specializing in resolution often are limited in range. High precision compact actuators do exist, such as models using light encoder and ultrasonic motors (SmarAct, Xeryon). However their costs are very high.

In addition, open-loop stages are almost never accurate, and the error term can shift over time and between different experimental configurations. The software readout of positions of an open-loop stage cannot be trusted to calculate deflections at the small scale of $1\mu\text{m}$. Even if we have highly precise stages, capacitance or interferometry based close-loop controls are still required for experiment-to-experiment consistency.

For our application in the live embryo, we consider motors with a range between $100\mu\text{m}$ and 1mm . Variable speed is important as the complex biological tissues show different behaviours at different strain rates. We also hope to multiplex (double in the current version) requiring small-sized motors. The piezoelectric benders have excellent resolution and accuracy given its crystal based mechanism, can handle a large range of frequencies, are small and cost-effective. They do have some drawbacks, as detected by the interferometer there is a residual vibration after movement, and the movement trajectory is not identical on voltage up and voltage down directions. The interferometer, while an overkill compared to resolution limit of the whole system (set by the probe-tip imaging to $\sim 1\mu\text{m}$), ensures precision on the holder detection. The current design leaves room for cellular level experiments once we can upgrade the base microscope.

Another possibility is to employ self-detecting probes where the electric conductivity inside the probe can be correlated to the deflection. These probes however are larger and stiffer and the signal could be affected by the wet tissue environment.

In the manuscript, we indicated that for lower-resolution and slower applications (Simple TiFM), a calibration between the probe position and the applied voltage will be sufficient, without the need for the interferometer.

4. Regarding the experiment in which the anterior and posterior PSM are stretched:

- Can the authors provide the movie ? that would allow the reader to visually compare the different responses?

Yes. The movie is now provided in the revision, Movie S2.

- Were posterior and anterior measurements performed in the same embryo?

Yes. We performed a paired measurement on each embryo.

- Because the authors measure strains and not mechanical properties I suggest that they be more cautious when claiming that "These results show that the aPSM is stiffer and more viscous than the pPSM" .

Thanks for the suggestion. We have now added more samples, model fitting and revised wording to more accurately describe the results.

- Regarding Figure 2B. What is the variability between embryos? Can the authors show the data for all 5 experiments in the supplementary? Why the imposed displacement steps don't have the same duration between aPMS and pPSM? While neither of them really reach asymptote? With equal duration it would be easier to compare their response.

In the **New Figure 2D-H**. We fitted the deformation recovery phase of the PSM across multiple embryos after probe retrieval using a two-term exponential model which captured the fast (1-5 seconds) and the slow (20-50 seconds) timescales. The model contains a residual strain term that is not accounted for by the generalized SLS model as a result of tissue plasticity. The results show variability of max and residual strains across embryos, yet consistently higher for the posterior PSM than the anterior PSM. The slow relaxation timescale is smaller for the posterior PSM, suggesting a more compliant tissue. These simple fittings provide useful quantifications on the differences

between tissue locations, but also show that simple viscoelastic models are insufficient to capture all aspects of the tissue behaviour observed.

There are small differences in the steps between runs as the command was entered manually. The duration at maximum is the same towards probe retrieval and is kept the same between different embryos. It is true that asymptote may not exist, as the tissues relax the loading stress they could continue to undergo viscous deformation. This can be seen in our long-term experiments (**New Figure 5H-I**).

- Regarding Figure 2C. The authors mention that "Upon probe retrieval after 5 minutes of holding, the tissues (now tracked manually by the insertion wound sites) both retracted, with the aPSM shortening faster and to a larger extent over time". This is not obvious from the figure, representing 1 experiment (?). How variable is the result from embryo to embryo? Is the difference significant? Can the authors fit a simple model, obtain the initial relaxation speed and do statistical tests using this? Again, an average or a plot showing all Ns would allow the reader to appreciate how reproducible the measurements are, especially since mechanical measurements are notoriously variable from embryo to embryo.

Thanks for these suggestions which we agree with fully. In the **New Figure 2D-H**. We fitted the deformation recovery phase of the PSM across multiple embryos after probe retrieval using a two-term exponential model which captured the fast (1-5 seconds) and the slow (20-50 seconds) timescales. The model contains a residual strain term that is not accounted for by the generalized SLS model as a result of tissue plasticity. The results show variability of max and residual strains across embryos, yet consistently higher for the posterior PSM than the anterior PSM. The slow relaxation timescale is smaller for the posterior PSM, suggesting a more compliant tissue. These simple fittings provide useful quantifications on the differences between tissue locations, but also show that simple viscoelastic models are insufficient to capture all aspects of the tissue behaviour observed.

Mechanical measurements are indeed variable as the reviewer pointed out! The 5 samples show variable differences between aPSM and pPSM (**New Figure 2E**). Part of these variabilities could be of technical origin (such as probe insertion depth), not necessarily all biological. We describe practices for minimizing the technical variabilities of the experiments in the revision.

5. Regarding Figure 3A :

- What do the authors mean when they mention " that the probes are calibrated through synchronized control"? is it in air, without resistance?
- P(R) and P(L) are not defined.

Yes, we run the compressive and extensive sequence first in the air or water to obtain the no-load gap between the 2 probes. This is clarified.

In the **new Figure 4A**, we provide a clear schematic for this experiment, and annotated the right probe P(R) and the left probe P(L).

6. Regarding Figure 4:

- The term "retraction" of the probe is not defined, I understand from the figure that it refers to decreasing the voltage, returning the holder to its initial position. But this should be defined as it could be confused with the "retrieval" of the probe, as in Fig2.

Thanks a lot for this. Retraction can be confusing to use. We now clarified that taking the probe out of the sample as "retrieval" throughout the manuscript. Moving the probe back from a stretched position is now called "return".

- The authors introduce the notion of plastic deformation, can they define it in particular in relation to viscous deformation?

Yes. Certain tissues (e.g., posterior body axis along the AP direction) under load show a small component of creep, yet the recoil after probe retrieval leaves a large residual deformation, suggesting a plastic component unaccounted for by the elastic and viscous components.

7. Regarding data in figure 5, axes in Figure 5F are not labeled.

They are now labelled in the **New Figure 6E-F**.

8. Regarding Data in Figure 6:

- Is the probe vertical, as shown in Figure 6A? if so this should be mentioned in the text. Are the two probes in Figure 6H also vertical? Is the calibration with no resistance performed prior to measurement? or there is no calibration anymore?

The single probe is largely vertical, usually with a small angle of +/- 3 degrees. The double probe setting is not vertical, as shown in Figure 1B(ii). The **new Figure 7A** provides a clear schematic for the double-probe rheology experiment.

In both single and double probe cases, we performed control/calibration experiments with no resistance in air or water. In this Figure only probe vibrational amplitudes were shown. We have not modelled the stress dynamics with the interferometer data in these experiments, given that the strain field and stress tensor modeling in these can be complex.

In the **New Figures 7A-C**. To test TiFM's capacity in mechanical property measurements, we performed new experiments with different concentrations of the standard viscoelastic material PIB. These materials were control-tested by a rheometer (**New Figure S1I**). The control values allowed us to compare it with the TiFM probe amplitudes generated both locally and across the sample at the second probe. We found that in a given setting (probe stiffness, depth and frequency) the probe amplitudes consistently and sensitively detect the differences across 3 orders of magnitude of the storage modulus of the material (7B(ii)) and are also sensitive to the frequency domain (7C(ii)). These results, together with the solid (hydrogel) results (**Figure 7D**), demonstrate that the probes have the resolution needed to measure the embryonic tissue properties which fall within this range. *While one may be tempted to use these data as calibration and then assign a modulus to the in vivo tissue measurements we performed, we refrain from doing that as the heterogeneity of small-tissue embryo environment is fundamentally different from simple polymer materials.*

We are currently creating a COMSOL model incorporating the probe geometry and measurement dynamics to recapitulate the measurement of complex modulus in control materials, as the first step. We will then extend this model by supplying inhomogeneous samples, such as two connected but distinct tissues (e.g., PSM and neural tube) where assumptions are made about their material properties and their ECM connections. These will then be fit with TiFM experimental data in and between these tissues. These steps are significant work and non-trivial, not possible to achieve in the current revision time frame.

Related to this point, a recent preprint (<https://doi.org/10.1101/2025.05.11.653307>) by the groups of Francis Corson and Jerome Gros presented a finite element model to interpret the mechanical properties of the quail blastoderm under a cantilever-driven deformation with an adhesive bead. This work sets up a great example of data fitting that we are learning from. The reviewers can gauge the work involved in the strain field measurement and the model fitting process in this work.

Significant changes to the modeling will be required for each of the versatile experimental settings presented in our work.

The above points are discussed in the revision.

- I have a similar comment that the one in point 2. Why don't the authors display the calibrated deflection of the cantilever rather than the position of chip and Tip? Again, aren't calibrated deflections the most informative information?

In these results we report the control/calibration amplitudes alongside the test groups. An alternative, as the reviewer suggested, is to calculate the maximum deflection against the control amplitudes, and to combine it with the probe stiffness and sample contact area to report maximum

probe stress. The current method has 3 advantages: firstly it provides an intuitive match with experimental images and movies (the stresses will be the opposite, smaller in larger amplitudes); secondly it offers a simple comparison between different sample groups (there are 6 different experiments/samples in Figure 7) without bringing errors in additional terms (insertion depth, angle, etc) in the calculation which may vary between experiment types; thirdly showing the control/calibration data allows readers to assess the baseline variability and error ranges of the system.

We are working on models that properly interpret the rheological properties of the samples and new probe head designs that will improve the interpretation of the deflection dynamics, to be reported in future versions of the system.

- The phase shift is not obvious in Figure 6B.

An example of larger phase shift is provided in the **new Figure S1J**, using a more viscous sample (high percentage PIB).

- There is no conclusion drawn from the result from figure 6C, regarding the relation between magnitude depth and frequency.

That's correct. We noted differences but it's not exactly clear how to interpret the quantitative difference depending on depth and frequency. With the addition of new samples for PIB at higher concentrations (**new Figure 7B-C**) and the hydrogel depth experiment (**Figure 7D**), we find this experiment not particularly informative and has moved it to the supplemental figure (Figure S1H).

- In Figure 6E, can the authors show here again the calibrated deflection instead of only the displacement of the tip?

This panel is intended to show the more variable tip tracking data in embryo tissues and how we fit them with a sine curve to obtain the amplitude. The deflection can be seen in the following panel (**Figure 7F**) as the amplitude difference between the no-load group (water) and the tissue groups.

9. Regarding Figure 7

- Same comments for previous figures, regarding the plotting of the calibrated deflection. It is unclear if the interferometer is still even used in these experiments.

Similarly, here the comparison is made between the control and different tissue locations, and now repeated across multiple samples in the **new Figure 8D-F**. In these particular zebrafish experiments, interferometer was not used as we found the low-speed piezo approaches to the samples used here are consistent between runs.

Reviewer 3: SUMMARY OF THE ADVANCE MADE IN THIS PAPER AND ITS POTENTIAL SIGNIFICANCE TO THE FIELD

The manuscript presents TiFM2.0, an upgraded tissue force microscopy system designed for live embryonic mechanical measurements. By incorporating interferometer positioning and a dual-probe configuration, the system allows for improved imaging accessibility and a wider range of mechanical perturbations, including bidirectional stretching, compression, and stress propagation experiments. The authors provide a set of proof-of-concept applications in chicken and zebrafish embryos, demonstrating the system's capability to measure and manipulate tissue deformation with precision. They also include simplified protocols to facilitate the system's adoption in other developmental biology laboratories.

The study looks technically robust, and the advancements over TiFM1.0 are clearly articulated. The ability to perform real-time, minimally invasive force measurements is a significant contribution to the field of morphogenesis and developmental mechanics. However, while the method is presented as a tool for stress and force measurements, there is no presentation of actual force-stress data, nor are the tissue deformations measured interpreted in terms of mechanical parameters (e.g., force vs. displacement curves, dynamic moduli). Additionally, the interpretation of stress as a

scalar rather than a tensor requires clarification. Finally, the manuscript lacks a quantitative assessment of inherent system errors and calibration uncertainties, and insufficient statistical analysis of the presented data.

We thank this reviewer for their thorough, critical and constructive comments.

SUGGESTIONS TO AUTHORS

(1) Lack of error estimation and calibration uncertainty

While the authors introduce interferometer positioning for improved precision, the manuscript does not provide a quantitative estimation of errors inherent to the system or potential calibration uncertainties. Addressing these issues is critical, as small variations in probe alignment, positioning drift, or sample deformation artifacts may introduce measurement errors.

We agree. In the revision we provide data and a discussion on different types of errors and uncertainties, so the readers are well-informed of the limitations of the system and can design and interpret experiments properly.

Specifically, In the revision we provide a detailed quantification and discussion on measurement errors, their origin, context and mitigation in the Materials & Methods section and refers to the key points in the Discussion.

Suggested improvements:

- Provide an explicit quantification of the measurement errors associated with probe positioning, force estimation, and strain measurements.

Probe positioning: The dominant error term sits at the resolution limit of the system, namely the measurement of the position of the probe tip (X_T). We mostly use ~30FPS imaging for most samples with a fast USB camera (up to ~100FPS). The most used 10x objective yields a pixel size of ~0.6 μ m (the thickness of the probe tips is 1-2 μ m). An algorithm detects and tracks the position of the tip at the pixel of sharpest gradient. This analysis pipeline has higher errors at faster probe movements. Errors also increase with imaging depth through tissue samples, as scattering deteriorates tip contrast and signal-to-noise ratio. In the small, multi-tissue embryos, tissue heterogeneity further complicates scattering. Fluorescently labelled probes can reduce errors, another option is to surgically remove overlying tissues to improve imaging clarity. In some cases, manual probe tracking was performed (Figure 5, 6B, 6C, 6H, 6I). At high frequencies probe positioning has an up-to +/- 4.8 μ m error range, this decreases to +/- 2.1 μ m at lower frequencies. In thick tissues we record a standard error in tracking of +/- 2.2 μ m across a variety of thick samples (e.g., somites, closed neural tube), this decreases to +/- 1.1 μ m in thinner tissues where the probe can be focused more clearly.

Piezo movement: Probe positioning is also dependant on voltage output, that determines the extent of piezo curvature. When manually setting the voltage using the MDT69XB interface, the outputted voltage deviates by ± 0.012 V. This is completely resolved by setting the voltage computationally using the instrument driver. In addition, the distance the piezo moves show some variation, with small voltage steps (3V) showing a standard deviation of +/- 0.22 μ m, whilst larger voltages have a greater standard deviation of +/- 3 μ m. For these reasons, the accurate tracking of the piezo's movement, using the interferometer, allows for the correction of these errors when using the data for force estimation.

Force estimation: Hardware components also carry uncertainties on their properties, for example, the manufacturer calibrated AFM cantilevers spring constants are accurate to a certain range. Insertion depth and modifications such as the epoxy-dye may affect the spring constants. Where necessary, manufacturer-individually-calibrated AFM cantilevers have been used. Insertion depth is precisely measured by identifying the top of the sample upon probe contact, the desired focal plane is then located and the change in focal plane records the insertion depth. This method results in insertion depth standard deviation of +/- 4 μ m.

Strain measurements: The strain measurements show errors of the same range as probe positioning as they have similar sources of error in image segmentation and tracking. The strain measurements may have larger errors in absolute terms in tissue samples as markers in tissues are not as high-contrast and may change over time. In relative terms, the errors in strain have a much smaller impact on mechanical property estimate, which is most sensitive to the probe positioning errors.

- Conduct control tests of the system using well-characterized synthetic material with known viscoelastic properties.

In the **New Figures 7A-C**, we performed new experiments with different concentrations of the standard viscoelastic material PIB. These materials were control-tested by a rheometer (**New Figure S11**). The control values allowed us to compare it with the TiFM probe amplitudes generated both locally and across the sample at the second probe. We found that in a given setting (probe stiffness, depth and frequency) the probe amplitudes consistently and sensitively detect the differences across 3 orders of magnitude of the storage modulus of the material (7B(ii)) and are also sensitive to the frequency domain (7C(ii)). These results, together with the solid (hydrogel) results (**Figure 7D**), demonstrate that the probes have the resolution needed to measure the embryonic tissue properties which fall within this range. *While one may be tempted to use these data as calibration and then assign a modulus to the in vivo tissue measurements we performed, we refrain from doing that as the heterogeneity of small-tissue embryo environment is fundamentally different from simple polymer materials.*

We are currently creating a COMSOL model incorporating the probe geometry and measurement dynamics to recapitulate the measurement of complex modulus in control materials, as the first step. We will then extend this model by supplying inhomogeneous samples, such as two connected but distinct tissues (e.g., PSM and neural tube) where assumptions are made about their material properties and their ECM connections. These will then be fit with TiFM experimental data in and between these tissues. These steps are significant work and non-trivial, not possible to achieve in the current revision time frame.

Related to this point, a recent preprint (<https://doi.org/10.1101/2025.05.11.653307>) by the groups of Francis Corson and Jerome Gros presented a finite element model to interpret the mechanical properties of the quail blastoderm under a cantilever-driven deformation with an adhesive bead. This work sets up a great example of data fitting that we are learning from. The reviewers can gauge the work involved in the strain field measurement and the model fitting process in this work. Significant changes to the modeling will be required for each of the versatile experimental settings presented in our work.

The above points are discussed in the revision.

- Discuss potential sources of noise (e.g., environmental factors, humidity sensitivity of capacitors in TiFM1.0) and how they have been mitigated in TiFM2.0.

Technical discussion on the sources of noise and comparisons between the two generations has been included in the Materials & Methods section where appropriate. We have not systematically measured the humidity sensitivity of capacitors in TiFM1.0. The observation was that condensations over time alter the capacitance readings while no piezo movement was instructed.

(2) Insufficient presentation of force or stress measurements

Although the manuscript presents TiFM2.0 as a mechanical measurement tool, there is a lack of explicit force or stress data: unless I missed a plot, all figures present measured displacements but never their translation into mechanically explicit information (which generally requires an explicit force-displacement measure and a mechanical model), such as Young's modulus, viscoelastic relaxation timescales, storage or loss moduli. The authors describe mechanical perturbations and tissue responses qualitatively but do not provide direct force-displacement or stress-strain curves.

Thanks! In the revision we added mechanically explicit information where applicable. We also remain cautionary on assigning physical quantities to the embryonic tissue locations, as discussed in more detail below, the complexity of the in vivo multi-tissue environment would require proper

models and a large amount of TiFM mapping data to properly fit for the moduli, which are beyond the scope of the example applications presented in the current manuscript.

Specific concerns:

- The stress formulation presented, $\sigma = k \cos(\theta) [f(x_{C1}) - x_{T1}] w^{-1} (z - z_0)^{-1}$, suggests a scalar representation rather than a full stress tensor. However, biological tissues exhibit anisotropic and spatially heterogeneous mechanical properties.

Yes. This quantity describes only the normal stress on the probe. As the probe moves in different tissues, it will detect different levels of this stress. By driving the probe in different directions in the tissue, we can also assess the spatial anisotropy (such as from between the A-P and M-L directions of the body axis, **Figure 5**). By performing a twisting experiment with double probes, we can begin to assess shear properties in the tissue. The current probe stress measurement by itself is not sufficient to reconstruct the stress tensor imposed by the probe or produced by tissues, or the anisotropic and spatially heterogeneous mechanical properties, and we do not claim so in the manuscript. The relative differences between different tissues, tissue locations and different directions from the simple measures (normal stress on the probe rather than the stress tensor, and tissue deformation along the probe movement direction rather than the strain field) remain interesting biological information. We have made this clear in the revision as follows:

“The normal force on the probe along the x axis at any time point is then given a by $F = k \cos(\theta) [f(x_{C1}) - x_{T1}]$ (Figure 1G, S1C,F), and can be converted to stress as $\sigma = Fw^{-1}(z-z_0)^{-1}$, where k is the spring constant, w is the width of the probes, z is the insertion depth, z_0 is the upper tissue surface, and θ is the insertion angle relative to the z axis. Note that under the embedded configuration of the probe-tissue interface, the detected force/stress on the probe has multiple sources of origin (such as compressive deformation along the x axis and shear deformation along the y axis near the probe edge). Therefore, while the results are indicative of the forces and material properties of the tissue location, appropriate models (e.g., finite element models) are needed to quantitatively interpret them together with the strain field around the probe measured by tracking markers and features from the images, to reveal mechanical heterogeneities and anisotropy of the sample (further discussed in Discussion section).”

Towards fine-grained, comprehensive results that can solve the anisotropic and heterogeneous properties of biological tissues as the reviewer point out, our next steps will include constructing higher resolution base microscopes to better map the strain fields in the samples, developing proper 2D/3D models and modifying probe tips accordingly (such as making it a small sphere to make the theory simpler), and fitting these with many data points in different directions, depth, and frequencies. We plan to first carry out these works in focused studies on specific tissues of interest (where certain simplifying assumptions could be made), together with molecular and embryological perturbations. These will be reported in future manuscripts.

- There seems not to be clear distinction between force and stress measurements, which could lead to misinterpretation of the mechanical properties being assessed.

We have taken care to make sure the stress calculation (normal stress only) is explained where applicable. The difference is when we divide the measured force by the probe-sample contact area, which we can calculate from the probe depth and probe width specifications.

Suggested improvements:

- Provide explicit force-displacement curves for different tissue types to illustrate mechanical responses under stretching and compression.

In the revision we provide examples of the force-displacement curves/coefficients.

New Figure 1E-G, S1A-F. In panels 1G,S1C,F we show the force dynamics in these samples (body axis tissues), the tissue resistance (Force vs Displacement) shown is in the linear regime, and a relaxation can be seen after each step-up of displacement. We have an ongoing project measuring the resistance, relaxation and limit of the linear regime (when the tissues yield) under different conditions such as signalling and extracellular matrix perturbations. These will be reported in future.

New Figure 4E-G (previous Figure 3). We show 3 neural tube samples under the compression and stretching by the double probe, and fitted the linear regimes to obtain the slope (Force vs Gap size, $\mu\text{N}/\mu\text{m}$ or N/m) as a simple way to relate neural tube shape (how open it is) to the measured force. The results show that the neural folds are more resistant in the compressive (lateral to medial) direction. Note that these quantities do not represent the stiffness of the neural tissue, as they come from a collective of factors such as tissue stiffness, shape, connections with neighbouring tissues, etc. We are modeling these factors and testing their effects in regulating the folding forces and resistances in a focused project on neural tube morphogenesis, to be reported in future.

- If possible, introduce stress-strain relationships or at least discuss why such characterization was not performed.

We agree with the reviewer that the stress-strain relationship will be important. Our primary caution here is that in our configuration, the probes are inserted into and surrounded by the sample, so the strain profile is not trivial. In Figure 2, we used a simplified measure of the strain (the tissue length between the 2 probes), which is far short of capturing the strain fields around the probes. For example, compressive strains and shear strains that contribute to the resistance to the probe are not measured.

As a test of the discrepancy, we performed similar stretching tests with probes inserted in $\sim 1\text{kPa}$ hydrogels. The calculated normal stress with the simple measure of the strain (the gel length between the 2 probes) at -2.4% is $\sim 500\text{Pa}$, however, the probes detected $\sim 2000\text{Pa}$, suggesting contributions from the surrounding gel e.g. shear strain. We plan to develop reliable quantitative models with these simple materials first, then move on to complex tissues with fitting. These will be future work.

These points are discussed in the revision.

- Clarify whether the force measurements correspond to local point forces or averaged tissue stresses, and justify the chosen approach.

Force measurements are recorded dynamically at the local contact point. This data can be used to obtain time-averaged stresses. Spatially, there is a certain interface size at the contact point depending on the type of probes and modifications used, and is usually larger than the size of several cells, meaning that the measurement is on the tissue scale stresses. In TiFM1.0 (Chan et al., 2023), we have shown continued tissue stress measurement with a single probe modified with a foil that matches the tissue cross-section, recording the initial stress and the stalling stress (when the tissue stops advancing against the probe). The general approach to measure average tissue stress will be to tailor and position the probe-tissue contact area to the area that the tissue stress acts on, this will be the quantity that effectively deforms the tissue. Small-scale local point forces may be less relevant on the tissue deformation questions.

The locality in terms of the origin of tissue stresses is less straightforward. For example, in the neural fold experiment (Figure 4), the tissue has a fold shape extended through the A-P axis. The measured forces at the probe contact point would have non-local contributions from the shape of the tissue.

- Discuss whether tensorial stress components could be extracted in future studies.

By driving the probe in different directions in the tissue, we can also assess the spatial anisotropy (such as from between the A-P and M-L directions of the body axis, Figure 5). By performing a twisting experiment with double probes, we can begin to assess shear properties in the tissue. The current probe stress measurement by itself is not sufficient to reconstruct the stress tensor

imposed by the probe or produced by tissues, or the anisotropic and spatially heterogeneous mechanical properties, and we do not claim so in the manuscript. The relative differences between different tissues, tissue locations and different directions from the simple measures (normal stress on the probe rather than the stress tensor, and tissue deformation along the probe movement direction rather than the strain field) remain interesting biological information.

Towards fine-grained, comprehensive results that can solve the anisotropic and heterogeneous properties of biological tissues as the reviewer point out, our next steps will include constructing higher resolution base microscopes to better map the strain fields in the samples, developing proper 2D/3D models and modifying probe tips accordingly (such as making it a small sphere to make the theory simpler), and fitting these with many data points in different directions, depth, and frequencies. We plan to first carry out these works in focused studies on specific tissues of interest (where certain simplifying assumptions could be made), together with molecular and embryological perturbations.

In addition, we can consider alternative motors and probes, such as rotational ones, to measure other degrees of freedom present in the tissue location.

A pathway towards this goal has been discussed in the revision

(3) Limited quantitative analysis of mechanical properties

(3.1) Limited quantitative analysis of measurements

The paper would benefit from a more extensive mechanical characterization of tissues beyond qualitative descriptions of deformation. Standard rheological measurements, such as storage (G') and loss (G'') moduli in oscillatory forcing experiments, are absent (Fig. 6).

This is our long-term aim, and TiFM has the potential to achieve the equivalent of standard rheological measurements in embryonic tissues (now supported by new data in **New Figure 7B,C**). At this stage however, significant work remains to model the inserted mode of measurement and the multi-scale, multi-component, multi-tissue local environments of the embryo properly. These are explained in detail below.

Suggested improvements:

- Provide at minimum viscoelastic timescale estimations by fitting relaxation curves with single or double exponential functions (Fig. 1E-F, Fig. 2B, Fig. 3A, Fig. 4A-C).

Thank you. We made the following changes:

New Figure 1E-G. We added panel G to show the force dynamics in this illustrative sample, the tissue resistance (Force vs Displacement) is in the linear regime, and a relaxation can be seen after each step-up of displacement.

New Figure 2D-H. We fitted the deformation recovery phase of the PSM across multiple embryos after probe retrieval using a two-term exponential model which captured the fast (1-5 seconds) and the slow (20-50 seconds) timescales. The model contains a residual strain term that is not accounted for by the generalized SLS model as a result of tissue plasticity. The results show variability of max and residual strains across embryos, yet consistently higher for the posterior PSM than the anterior PSM. The slow relaxation timescale is smaller for the posterior PSM, suggesting a more compliant tissue. These simple fittings provide useful quantifications on the differences between tissue locations, but also show that simple viscoelastic models are insufficient to capture all aspects of the tissue behaviour observed.

New Figure 4E-G (previous Figure 3). We show 3 neural tube samples under the compression and stretching by the double probe, and fitted the linear regimes to obtain the slope (Force vs Gap size) as a simple way to relate neural tube shape (how open it is) to the measured force. The results show that the neural folds are more resistant in the compressive (lateral to medial) direction. Note that these quantities do not represent the stiffness of the neural tissue, as they come from a collective of factors such as tissue stiffness, shape, connections with neighbouring tissues, etc.

New Figure 5C-G (previous Figure 4). We added multiple embryos and tracked their deformation and recovery. The recovery kymographs were fitted with an exponential term and the tissue displacements and the timescale of recovery were compared across embryos and close and far tissue locations. The results show an enhanced deformability along the AP axis at the posterior progenitor domain near the compressive probe, but not along the ML axis.

- In Fig. 6, perform oscillatory mechanical testing at varying frequencies to determine the full spectral viscoelastic response of embryonic tissues, extracting G' and G'' as a function of forcing frequency.

In the revision we show an example of oscillatory mechanical testing with varying frequencies in a calibrated PIB preparation (**new Figure 7B,C(ii)**). These data will be used to test future models relating probe geometry, stiffness and movement to the G' and G'' of the sample, as we move towards doing the same for embryonic tissues.

This exact size of tissue that is impacted by the probe has implications on data interpretation (e.g., determining the strain). It is not immediately clear how far the mechanical impact of the probe reaches in the complex *in vivo*, *in situ* context. The size of probed regions in simple materials such as the hydrogel and PIB can be defined by tracker particles or the second probe. We have results of hydrogel deformations (loaded with tracker beads and analyzed by PIV) under the oscillatory probe but the data is yet to be 3D. Similarly, in the embryonic tissue, comprehensive imaging will be required to map the strain fields around the probe. In a solid tissue like the notochord, the actuation could propagate far along the notochord. In a fluid tissue like the posterior PSM, the actuation may drop quickly near the probe (e.g., Figure 7H). Still, as the tissue mechanical properties are complex with multiple structural components, an elastic component could propagate a small part of the probe actuation in the posterior PSM to far distances. We are actively investigating these questions by performing systematic mapping with the TiFM. The findings will be reported in future manuscripts.

The full spectral viscoelastic response of embryonic tissues is a certainly our long-term goal. We are working on variable frequency tests in embryonic tissues, and can drive probes using frequency informed signals such as the “Chirp” input signal with a frequency sweep (Geri et al., 2018). However, the complexity of the strain field which is difficult to quantify with the current imaging set-up, and the multi-tissue environment and potential non-local contributors to the probe movement profile, make it out of scope at this stage to correctly capture and model the strain dynamics and the stress tensor imposed on the tissue. Applying grossly simplified fit to obtain G' and G'' may lead to erroneous and misleading results.

- Compare measured forces and mechanical moduli with some known values from previous studies (relates to Point 1, control tests).

In the revision we compare the results with existing information whenever applicable, and with control material results. Different methods have different caveats. For these embryonic tissues, there are some reported values using pipette aspiration, larger-scale indenters, AFM, magnetic droplet, etc. There is a reasonable amount of agreement on relative differences (such as notochord is stiffer than the PSM). The “known” values vary across a few orders of magnitude. Our results are not outside this range.

(3.2) Insufficient statistical analysis

While the study presents measurements across different conditions, the statistical analysis is limited, and the manuscript lacks a detailed quantification of measurement variability across biological replicates. This is particularly critical for a biomechanical study, where sample-to-sample variation could be significant.

We strongly agree and included additional samples and statistics where appropriate.

Suggested improvements:

- Report mean values with standard deviations or confidence intervals for key measured mechanical parameters (points 2 and 3.1).
- Provide sample sizes (N) for each measurement condition and include statistical tests.

New Figure 1. Illustration is improved and a clearer representative example of measurement principle is included (with additional replicates shown in **New Figure S1**). No biological conclusion is made here.

New Figure 2. D-H, a group of embryos were fitted and their fast and slow relaxation timescales were compared.

New Figure 3. This expands the analysis of cellular changes under the TiFM double probe stretching experiment. A large number of cells were segmented and compared between 2 regions of the movie. Cell shapes and orientations were compared systematically with statistics. This movie and analysis are representative of other repeats.

New Figure 4. Additional samples added and the resistance in compressive and extensive directions were compared.

New Figure 5. Additional samples added, fitted, and compared several metrics statistically.

New Figure 6. Presentation is improved with labels. Variability of blood pressure estimation is noted in the main text.

New Figure 7. New PIB sample added with statistical tests. All results in this Figure now have multiple samples with statistics.

New Figure 8. Additional samples added and the results statistically analyzed. Sample number N is noted in all legends and texts where applicable.

Minor comments:

- Figure 1: The schematic of the TiFM2.0 setup could be enhanced by providing a clearer comparison to TiFM1.0, highlighting the key technical improvements.

We added a schematic of the single arm in the **new Figure 2B(i)** to compare with the double arm configuration. We also added a **new Figure 2B(iii)** to illustrate the mechanism of the interferometry.

- Figure 2: Tissue stretching experiments would be strengthened by additional kinematic tracking of cell shape deformations.

The **New Figure 3** provides a comprehensive cell shape analysis and tracking, to strengthen the demonstration example. Further work on this (including junctional dynamics, cytoskeleton changes, tissue fluidity changes) will be reported in a standalone manuscript in preparation.

- The term "stress" should be used carefully to avoid confusion with "force," particularly when presenting data in biological contexts.

Certainly, we have taken care to make sure the stress calculation is explained where applicable. The difference is when we divide the measured force by the probe-sample contact area, which we can calculate from the probe depth and probe width specifications. We also paid attention to the wording when discussing tissue generated stresses and forces.

Conclusion

This study presents a valuable technological advancement in the field of live embryo mechanics. TiFM2.0 offers improved precision and versatility compared to its predecessor, and its ability to perform bidirectional perturbations is a significant step forward. However, the manuscript would benefit from a more rigorous error analysis, clearer distinction between stress and force measurements, and a more thorough analysis of tissue mechanical properties. Addressing these concerns could significantly enhance the impact and reproducibility of the findings

Thank you! We believe that in the revision we have provided a more thorough coverage of errors and variabilities, clarified terminology, and reported additional experiments to address reproducibility. We also included standard material controls that show the capacity of the system to measure tissue mechanical properties. However, we are not in a position yet to precisely translate our results to stress tensors the tissues experience/produce, or rheological modulus measurements that can be interpreted and compared to standard materials. Many challenges remain in the context of the complex multi-tissue environment of the small embryos, particularly in defining the strain fields and in identifying distinct contributors to the local mechanical response. The reviewer's comments help guide us to the right direction. We have in the revision made sure the manuscript clearly describes these limitations and future directions, and do not oversale the power of the technique in its current form. Nonetheless, our example applications demonstrated here are new and of interest to the developmental biology community.

Second decision letter

MS ID#: dev.204549R1

MS TITLE: TiFM2.0 - Versatile mechanical measurement and actuation in live embryos

AUTHORS: Ana R. Hernandez-Rodriguez, Yisha Lan, Fengtong Ji, Susannah B.P. McLaren, Joana M. N. Vidigueira, Ruoheng Li, Yixin Dai, Emily Holmes, Lauren D. Moon, Lakshmi Balasubramaniam and Fengzhu Xiong

Dear Dr Xiong,

I have now received all the referees reports on the above manuscript, and have reached a decision. The referees' comments are appended below.

The overall evaluation is positive and we would like to publish a revised manuscript in Development, provided that the referees' comments can be satisfactorily addressed. Please attend to all of the reviewers' comments in your revised manuscript and detail them in your point-by-point response. I draw your attention in particular to point 2 of Reviewer 2 and to the main comment from Reviewer 3 which have to be addressed. If you do not agree with any of their criticisms or suggestions explain clearly why this is so. If it would be helpful, you are welcome to contact us to discuss your revision in greater detail. Please send us a point-by-point response indicating your plans for addressing the referees' comments, and we will look over this and provide further guidance.

Reviewer 1

Advance summary and potential significance to field

I very much appreciate the clarifications, additional examples and analyses that have now been included in the revised manuscript. The main novelty of this work is to provide a novel setup design to perform tissue force microscopy in live tissues, using interferometer positioning and a dual probe design that now enables direct force actuation in situ. While I am convinced the approaches put forward here are valuable, I think the technology, protocols and data interpretation are not entirely mature for rapid adoption for new users and considerable work is clearly still ongoing to bring this approach fully to its ambition. Nonetheless, I appreciate the more detailed methods section and the more thorough discussion of the relevant sources of technical variability with this approach. Overall, this study is of interest to a broad audience and still represents an advance for the mechanobiology toolkit, especially for probing early embryonic tissues.

Comments for the author

The discussion now reflects the state and potential of this technology, as well as the required next steps. A few minor points can still be included in the discussion, namely: a) how the "exact size of tissue that is impacted by the probe has implications on data interpretation" - the authors discuss this point in response to point 4 and I believe this to be useful for future adopters; and, b) comment on the how the size of the probe versus cell size impacts the measurements.

Reviewer 2

Advance summary and potential significance to field

The authors have addressed most of my earlier concerns. In particular, they have clarified how the system operates and how the measurements are obtained, and they have added data from additional embryos. The authors are appropriately cautious in their conclusions regarding the mechanical properties of the tissues, which nevertheless results in valuable relative measurements.

I still believe the calibration data (e.g., Fig. 1E) are not controls and would be better placed in the Supplement and that the meaningful deflection is somehow hidden; Showing only the calibration in the main figures is somewhat misleading, but at least the revised text now clarifies this point.

I do have some issues about the new data:

1. Regarding the mechanical model in Figure 2D. How does adding a spring in parallel to the spring-dashpot elements account for residual strain? Intuitively, wouldn't persistent deformation be captured by a dashpot in series, not an added parallel spring?
2. All three reviewers requested an increase in sample size for most experiments, yet the added Figure 3D again shows an example from a single embryo. The authors report a highly significant difference in cell area/apsect ratio using a two-tailed Welch's t-test; however, this test was performed on hundreds of cells from a single embryo. Since all these measurements come from one biological specimen, the observations are not independent, and the test's assumptions are violated. The apparent significance is therefore not meaningful. To assess whether this effect is biologically reproducible, the analysis should be based on multiple embryos, treating each embryo as one replicate, or alternatively use a mixed-effects model that accounts for the nested structure of the data (cells within embryos). The same applies for the test performed on the 10 cells analyzed (size and angles) and everywhere data from a single embryo are analyzed and tests used. Furthermore, adding an an overlay of the masks on the raw images with cell outlines, would have been appreciated in order to show that cell segmentaion is indeed accurate.

Reviewer 4

Advance summary and potential significance to field

Overall, the authors have addressed the concerns reasonably well. However, my main concern is closely aligned with one previously raised by the referees. The authors do not provide absolute measurements or estimates of viscoelastic parameters. I agree that, as they mention, "creating a COMSOL model incorporating the probe geometry and measurement dynamics to recapitulate the measurement of complex modulus in control materials" would be important for more accurate estimates in the future. Nevertheless, reasonable assumptions can still be made, and an estimate of the moduli can be provided. And in any case, I am not convinced that a finite element model (such as COMSOL or another) will be useful in addressing the potential issue of "heterogeneity in the small-tissue embryo environment" that they use as an argument for not making the estimates. I praise the authors for their experimental efforts in making careful measurements, and I believe they should not refrain from estimating parameters that are challenging to obtain without such an experimental approach.

Minor:

I find that the introduction would benefit from citing recent works on force inference and in situ quantitative measurements, which are relatively few. The review suggests that force inference has been performed only in 2D, whereas recent work has demonstrated their ability to infer tension in

3D (see, for example, DOI: 10.1038/s41592-023-02084-7). Similarly, the paper does not refer to quantitative measurements with non-contact methods (e.g. DOIs: 10.1073/pnas.1418732112, 10.1016/j.cub.2017.09.00, 10.1242/dev.175109), while they provide measurements of viscoelastic properties, which are directly relevant to the present manuscript.

Second revision

Author response to reviewers' comments

Reviewer 1: SUMMARY OF THE ADVANCE MADE IN THIS PAPER AND ITS POTENTIAL SIGNIFICANCE TO THE FIELD

I very much appreciate the clarifications, additional examples and analyses that have now been included in the revised manuscript. The main novelty of this work is to provide a novel setup design to perform tissue force microscopy in live tissues, using interferometer positioning and a dual probe design that now enables direct force actuation in situ. While I am convinced the approaches put forward here are valuable, I think the technology, protocols and data interpretation are not entirely mature for rapid adoption for new users and considerable work is clearly still ongoing to bring this approach fully to its ambition. Nonetheless, I appreciate the more detailed methods section and the more thorough discussion of the relevant sources of technical variability with this approach. Overall, this study is of interest to a broad audience and still represents an advance for the mechanobiology toolkit, especially for probing early embryonic tissues.

Thanks very much. We continue to refine and simplify this system for wider community access. The reviewer's feedback helps us identify and focus on key areas of improvement.

SUGGESTIONS TO AUTHORS

The discussion now reflects the state and potential of this technology, as well as the required next steps. A few minor points can still be included in the discussion, namely: a) how the "exact size of tissue that is impacted by the probe has implications on data interpretation" - the authors discuss this point in response to point 4 and I believe this to be useful for future adopters; and, b) comment on the how the size of the probe versus cell size impacts the measurements.

These points have been discussed accordingly. Specifically, for a), it reads in Discussion: "*Here it will be important to consider the property of the tissue that defines the range that the probes are detecting which need to be imaged. For example, a solid tissue like the notochord will propagate the deformation to far distances; whereas the PSM will dampen it near the actuation site. Still, even the most fluid-like pPSM propagates a fraction of deformation to far distances, showing the multilayered structural complexity of tissues that need to be taken into consideration.*" For b), it reads: "*cell sizes at these stages are on the order of 10 μm so the probes primarily measure tissue-level properties*".

Reviewer 2: The authors have addressed most of my earlier concerns. In particular, they have clarified how the system operates and how the measurements are obtained, and they have added data from additional embryos. The authors are appropriately cautious in their conclusions regarding the mechanical properties of the tissues, which nevertheless results in valuable relative measurements.

Thanks very much. We continue to develop data interpretation methods. The reviewer's feedback helps us identify and focus on key areas of improvement.

I still believe the calibration data (e.g., Fig. 1E) are not controls and would be better placed in the Supplement and that the meaningful deflection is somehow hidden; Showing only the calibration in the main figures is somewhat misleading, but at least the revised text now clarifies this point.

Yes “control” is a confusing label in this context and we have changed it to “calibration” for clarity. We also changed “sample” to “test” for Fig. 1F. This helps the precision of the presentation and we thank the reviewer for this point.

As a technical article we thought it would be good for the readers to get a sense of what the raw data look like (Fig.1E-F). Fig.1G (showing the force) is equivalent to a plot of deflection. This is further clarified in the text.

I do have some issues about the new data:

1. Regarding the mechanical model in Figure 2D. How does adding a spring in parallel to the spring-dashpot elements account for residual strain? Intuitively, wouldn't persistent deformation be captured by a dashpot in series, not an added parallel spring?

We found that the creep stops during the stretching experiment, suggesting a long-term spring (a series dashpot will predict continued yielding). In this situation, the residual strain would suggest a plastic deformation (the long-term parallel spring has energy dissipation and does not fully recoil). This way the presented diagram is closer to the experiment, and we discuss this plasticity in the text.

2. All three reviewers requested an increase in sample size for most experiments, yet the added Figure 3D again shows an example from a single embryo. The authors report a highly significant difference in cell area/aspect ratio using a two-tailed Welch's t-test; however, this test was performed on hundreds of cells from a single embryo. Since all these measurements come from one biological specimen, the observations are not independent, and the test's assumptions are violated. The apparent significance is therefore not meaningful. To assess whether this effect is biologically reproducible, the analysis should be based on multiple embryos, treating each embryo as one replicate, or alternatively use a mixed-effects model that accounts for the nested structure of the data (cells within embryos). The same applies for the test performed on the 10 cells analyzed (size and angles) and everywhere data from a single embryo are analyzed and tests used.

Thanks for pointing this out and we agree completely. The cells are stretched together in a connected tissue field and are not independently stretched from each other in a single embryo. We have removed the inappropriate statistics from the distribution of cell size or aspect ratio within a single embryo (3B-C), and just present the measured distributions as they are.

To check the effect of stretching on cell shapes, we added data from 2 more independent embryos. We took the population average of each embryo and pair-tested them (new 3E-F). The results indicate a directional change of cell size and aspect ratio in region 1 (between the probes) but not region 2 (away from the stretched region).

Similarly for the cell tracking results, we now describe them directly without tests, and show the observations are consistent in 2 more independent embryos (new S1G-L).

Furthermore, adding an an overlay of the masks on the raw images with cell outlines, would have been appreciated in order to show that cell segmentaion is indeed accurate.

We added a representative zoom-in of raw data and the corresponding masks (new Fig.3D').

Reviewer 4: Overall, the authors have addressed the concerns reasonably well. However, my main concern is closely aligned with one previously raised by the referees. The authors do not provide absolute measurements or estimates of viscoelastic parameters. I agree that, as they mention, "creating a COMSOL model incorporating the probe geometry and measurement dynamics to recapitulate the measurement of complex modulus in control materials" would be important for more accurate estimates in the future. Nevertheless, reasonable assumptions can still be made, and an estimate of the moduli can be provided. And in any case, I am not convinced that a finite element model (such as COMSOL or another) will be useful in addressing the potential issue of "heterogeneity in the small-tissue embryo environment" that they use as an argument for not making the estimates. I praise the authors for their experimental efforts in making careful measurements, and I believe they should not refrain from estimating parameters that are challenging to obtain without such an experimental approach.

Thanks a lot for the reviewer's comments. We'd appreciate if the reviewer could describe the specific limitations of our proposed future multi-tissue modelling approach, or suggest alternatives to finite element models to address the heterogeneity. We very much want to make the best estimates possible and inform the readers of the caveats. We have noted the reviewer's point that the model may not be successful in addressing the heterogeneity issue in the revision. Specifically,

in discussion: “How well this approach will be able to address the complex and heterogenous mechanical environment of the tissue remains to be explored.”

Following the reviewer’s suggestions, and given our tests on hydrogels and PIBs, we can make some cautious estimates on the viscoelastic properties of the embryonic tissue locations. In the revision, we provide estimates of PSM, NT and notochord. Specifically, in the results section of Fig.7, we added: “Assuming local homogeneity similar to control materials near the probe, we estimate the notochord and anterior NT stiffness to be in the 1-3 kPa range whereas the pPSM will be in the 0.1-0.4 kPa range. At 0.5 Hz, the storage modulus of the PSM tissue will be ~300 Pa, and shows a small but consistent difference between the aPSM and pPSM.”

Minor:

I find that the introduction would benefit from citing recent works on force inference and in situ quantitative measurements, which are relatively few. The review suggests that force inference has been performed only in 2D, whereas recent work has demonstrated their ability to infer tension in 3D (see, for example, DOI: 10.1038/s41592-023-02084-7). Similarly, the paper does not refer to quantitative measurements with non-contact methods (e.g. DOIs: 10.1073/pnas.1418732112, 10.1016/j.cub.2017.09.00, 10.1242/dev.175109), while they provide measurements of viscoelastic properties, which are directly relevant to the present manuscript.

Thank you. We have added these relevant references in the introduction paragraph. Optical tweezer is an important method to measure junctional forces and together with knowledge or assumption of the viscosity of the cytosol/cortex allows viscoelastic properties to be measured. This is on a smaller scale but the principle is of course relevant to discuss in this manuscript.

Third decision letter

MS ID#: dev.204549R2

MS TITLE: TiFM2.0 - Versatile mechanical measurement and actuation in live embryos

AUTHORS: Ana R. Hernandez-Rodriguez, Yisha Lan, Fengtong Ji, Susannah B.P. McLaren, Joana M. N. Vidigueira, Ruoheng Li, Yixin Dai, Emily Holmes, Lauren D. Moon, Lakshmi Balasubramaniam and Fengzhu Xiong

Dear Dr Xiong,

I am happy to tell you that your manuscript has been accepted for publication in Development, pending our standard publication integrity checks.

Reviewer 4

Advance summary and potential significance to field

The authors have adequately addressed my concerns.